# POINCARÉ GLOVE: HYPERBOLIC WORD EMBEDDINGS

**Alexandru Țifrea**[∗]**, Gary Bécigneul**[∗]**, Octavian-Eugen Ganea**[∗]
Department of Computer Science
ETH Zürich, Switzerland
`tifreaa@ethz.ch,{gary.becigneul,octavian.ganea}@inf.ethz.ch`

## ABSTRACT

Words are not created equal. In fact, they form an aristocratic graph with a latent hierarchical structure that the next generation of unsupervised learned word embeddings should reveal. In this paper, justified by the notion of delta-hyperbolicity or tree-likeliness of a space, we propose to embed words in a Cartesian product of hyperbolic spaces which we theoretically connect to the Gaussian word embeddings and their Fisher geometry. This connection allows us to introduce a novel principled hypernymy score for word embeddings. Moreover, we adapt the well-known Glove algorithm to learn unsupervised word embeddings in this type of Riemannian manifolds. We further explain how to solve the analogy task using the Riemannian parallel transport that generalizes vector arithmetics to this new type of geometry. Empirically, based on extensive experiments, we prove that our embeddings, trained *unsupervised*, are the first to simultaneously outperform strong and popular baselines on the tasks of similarity, analogy and hypernymy detection. In particular, for word hypernymy, we obtain new state-of-the-art on fully unsupervised WBLESS classification accuracy.

## 1 INTRODUCTION & MOTIVATION

Word embeddings are ubiquitous nowadays as first layers in neural network and deep learning models for natural language processing. They are essential in order to move from the discrete word space to the continuous space where differentiable loss functions can be optimized. The popular models of Glove (Pennington et al., 2014), Word2Vec (Mikolov et al., 2013b) or FastText (Bojanowski et al., 2016), provide efficient ways to learn word vectors fully unsupervised from raw text corpora, solely based on word co-occurrence statistics. These models are then successfully applied to word similarity and other downstream tasks and, surprisingly (or not (Arora et al., 2016)), exhibit a linear algebraic structure that is also useful to solve word analogy.

However, unsupervised word embeddings still largely suffer from revealing asymmetric word relations including the latent hierarchical structure of words. This is currently one of the key limitations in automatic text understanding, e.g. for tasks such as textual entailment (Bowman et al., 2015). To address this issue, (Vilnis & McCallum, 2015; Muzellec & Cuturi, 2018) propose to move from point embeddings to probability density functions, the simplest being Gaussian or Elliptical distributions. Their intuition is that the variance of such a distribution should encode the generality/specificity of the respective word. However, this method results in losing the arithmetic properties of point embeddings (e.g. for analogy reasoning) and becomes unclear how to properly use them in downstream tasks. To this end, we propose to take the best from both worlds: we embed words as points in a Cartesian product of hyperbolic spaces and, additionally, explain how they are bijectively mapped to Gaussian embeddings with diagonal covariance matrices, where the hyperbolic distance between two points becomes the Fisher distance between the corresponding probability distribution functions (PDFs). This allows us to derive a novel principled is-a score on top of word embeddings that can be leveraged for hypernymy detection. We learn these word embeddings unsupervised from raw text by generalizing the Glove method. Moreover, the linear arithmetic property used for solving word analogy has a mathematical grounded correspondence in this new space based on the established notion of parallel transport in Riemannian manifolds. In addition, these hyperbolic embeddings outperform Euclidean Glove on word similarity benchmarks. We thus describe, to our knowledge,

---

[∗]All authors contributed equally.

the first word embedding model that competitively addresses the above three tasks simultaneously. Finally, these word vectors can also be used in downstream tasks as explained by Ganea et al. (2018b).

We provide additional reasons for choosing the hyperbolic geometry to embed words. We explain the notion of average $\delta$-hyperbolicity of a graph, a geometric quantity that measures its "democracy" (Borassi et al., 2015). A small hyperbolicity constant implies "aristocracy", namely the existence of a small set of nodes that "influence" most of the paths in the graph. It is known that real-world graphs are mainly complex networks (e.g. scale-free exhibiting power-law node degree distributions) which in turn are better embedded in a tree-like space, i.e. hyperbolic (Krioukov et al., 2010). Since, intuitively, words form an "aristocratic" community (few generic ones from different topics and many more specific ones) and since a significant subset of them exhibits a hierarchical structure (e.g. WordNet (Miller et al., 1990)), it is naturally to learn hyperbolic word embeddings. Moreover, we empirically measure very low average $\delta$-hyperbolicity constants of some variants of the word log-co-occurrence graph (used by the Glove method), providing additional quantitative reasons for why spaces of negative curvature (i.e. hyperbolic) are better suited for word representations.

## 2 RELATED WORK

Recent *supervised methods* can be applied to embed any tree or directed acyclic graph in a low dimensional space with the aim of improving link prediction either by imposing a partial order in the embedding space (Vendrov et al., 2015; Vilnis et al., 2018; Athiwaratkun & Wilson, 2018), by using hyperbolic geometry (Nickel & Kiela, 2017; 2018), or both (Ganea et al., 2018a).

To learn *word embeddings* that exhibit hypernymy or hierarchical information, supervised methods (Vulić & Mrkšić, 2018; Nguyen et al., 2017) leverage external information (e.g. WordNet) together with raw text corpora. However, the same goal is also targeted by more ambitious fully unsupervised models which move away from the "point" assumption and learn various probability densities for each word (Vilnis & McCallum, 2015; Muzellec & Cuturi, 2018; Athiwaratkun & Wilson, 2017; Singh et al., 2018).

There have been two very recent attempts at learning unsupervised word embeddings in the hyperbolic space (Leimeister & Wilson, 2018; Dhingra et al., 2018). However, they suffer from either not being competitive on standard tasks in high dimensions, not showing the benefit of using hyperbolic spaces to model asymmetric relations, or not being trained on realistically large corpora. We address these problems and, moreover, the connection with density based methods is made explicit and leveraged to improve hypernymy detection.

## 3 HYPERBOLIC SPACES AND THEIR CARTESIAN PRODUCT

In order to work in the hyperbolic space, we have to choose one *model*, among the five isometric models that exist. We choose to embed words in the Poincaré ball $\mathbb{D}^n = \{x \in \mathbb{R}^n \mid \|x\|_2 < 1\}$. This is illustrated in Figure 1a for $n = 2$, where dark lines represent geodesics. The distance function in this space is given by $d_{\mathbb{D}^n}(x, y) = \cosh^{-1}\left(1 + \lambda_x \lambda_y \|x - y\|_2^2/2\right)$, $\lambda_x := 2/(1 - \|x\|_2^2)$ being the *conformal factor*. We will also embed words in products of

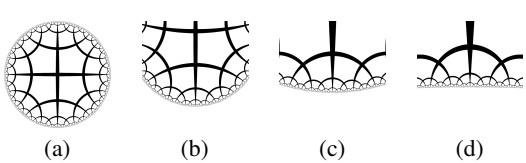

(a)  (b)  (c)  (d)

Figure 1: Isometric deformation $\varphi$ of $\mathbb{D}^2$ into $\mathbb{H}^2$.

hyperbolic spaces, and explain why later on. A product of $p$ balls $(\mathbb{D}^n)^p$, with the induced product geometry, is known to have distance function $d_{(\mathbb{D}^n)^p}(x, y)^2 = \sum_{i=1}^p d_{\mathbb{D}^n}(x_i, y_i)^2$. Finally, another model of interest for us is the Poincaré half-plane $\mathbb{H}^2 = \mathbb{R} \times \mathbb{R}_+^*$ illustrated in Figure 1d, with distance function $d_{\mathbb{H}^2}(x, y) = \cosh^{-1}\left(1 + \|x - y\|_2^2/(2y_1 y_2)\right)$. Figure 1 shows an isometry $\varphi$ from $\mathbb{D}^2$ to $\mathbb{H}^2$ mapping the vertical segment $\{0\} \times (-1, 1)$ to $\mathbb{R}_+^*$ and fixing $(0, 1)$, sending the radius to $\infty$.

## 4  ADAPTING GLOVE

**Euclidean GLOVE.**  The GLOVE (Pennington et al., 2014) algorithm is an unsupervised method for learning word representations in the Euclidean space from statistics of word co-occurrences in a text corpus, with the aim to geometrically capture the words' meaning and relations.

We use the notations: $X_{ij}$ is the number of times word $j$ occurs in the same window context as word $i$; $X_i = \sum_k X_{ik}$; $P_{ij} = X_{ij}/X_i$ is the probability that word $j$ appears in the context of word $i$. An embedding of a (target) word $i$ is written $w_i$, while an embedding of a context word $k$ is written $\tilde{w}_k$.

The initial formulation of the GLOVE model suggests to learn embeddings as to satisfy the equation $w_i^T \tilde{w}_k = \log(P_{ik}) = \log(X_{ik}) - \log(X_i)$. Since $X_{ik}$ is symmetric in $(i, k)$ but $P_{ik}$ is not, (Pennington et al., 2014) propose to restore the symmetry by introducing biases for each word, absorbing $\log(X_i)$ into $i$'s bias:

$$w_i^T \tilde{w}_k + b_i + \tilde{b}_k = \log(X_{ik}). \tag{1}$$

Finally, the authors suggest to enforce this equality by optimizing a weighted least-square loss:

$$J = \sum_{i,j=1}^V f(X_{ij}) \left( w_i^T \tilde{w}_j + b_i + \tilde{b}_j - \log X_{ij} \right)^2, \tag{2}$$

where $V$ is the size of the vocabulary and $f$ down-weights the signal coming from frequent words (it is typically chosen to be $f(x) = \min\{1, (x/x_m)^\alpha\}$, with $\alpha = 3/4$ and $x_m = 100$).

**GLOVE in metric spaces.**  Note that there is no clear correspondence of the Euclidean inner-product in a hyperbolic space. However, we are provided with a distance function. Further notice that one could rewrite Eq. (1) with the Euclidean distance as $-\frac{1}{2}\|w_i - \tilde{w}_k\|^2 + b_i + \tilde{b}_k = \log(X_{ik})$, where we absorbed the squared norms of the embeddings into the biases. We thus replace the GLOVE loss by:

$$J = \sum_{i,j=1}^V f(X_{ij}) \left( -h(d(w_i, \tilde{w}_j)) + b_i + \tilde{b}_j - \log X_{ij} \right)^2, \tag{3}$$

where $h$ is a function to be chosen as a hyperparameter of the model, and $d$ can be any differentiable distance function. Although the most direct correspondence with GLOVE would suggest $h(x) = x^2/2$, we sometimes obtained better results with other functions, such as $h = \cosh^2$ (see sections 8 & 9). Note that De Sa et al. (2018) also apply $\cosh$ to their distance matrix for hyperbolic MDS before applying PCA. Understanding why $h = \cosh^2$ is a good choice would be interesting future work.

## 5  CONNECTING GAUSSIAN EMBEDDINGS & HYPERBOLIC EMBEDDINGS

In order to endow Euclidean word embeddings with richer information, Vilnis & McCallum (2015) proposed to represent words as Gaussians, *i.e.* with a mean vector and a covariance matrix[1], expecting the variance parameters to capture how generic/specific a word is, and, hopefully, entailment relations. On the other hand, Nickel & Kiela (2017) proposed to embed words of the WordNet hierarchy (Miller et al., 1990) in hyperbolic space, because this space is mathematically known to be better suited to embed tree-like graphs. It is hence natural to wonder: is there a connection between the two?

**The Fisher geometry of Gaussians is hyperbolic.**  It turns out that there exists a striking connection (Costa et al., 2015). Note that a *1D Gaussian $\mathcal{N}(\mu, \sigma^2)$ can be represented as a point $(\mu, \sigma)$ in* $\mathbb{R} \times \mathbb{R}_+^*$. Then, the Fisher distance between two distributions relates to the hyperbolic distance in $\mathbb{H}^2$:

$$d_F\left(\mathcal{N}(\mu, \sigma^2), \mathcal{N}(\mu', \sigma'^2)\right) = \sqrt{2} d_{\mathbb{H}^2}\left((\mu/\sqrt{2}, \sigma), (\mu'/\sqrt{2}, \sigma')\right). \tag{4}$$

For $n$-dimensional Gaussians with diagonal covariance matrices written $\Sigma = \mathrm{diag}(\sigma)^2$, it becomes:

$$d_F\left(\mathcal{N}(\mu, \Sigma), \mathcal{N}(\mu', \Sigma')\right) = \sqrt{\sum_{i=1}^n 2 d_{\mathbb{H}^2}\left((\mu_i/\sqrt{2}, \sigma_i), (\mu_i'/\sqrt{2}, \sigma_i')\right)^2}. \tag{5}$$

Hence there is a direct correspondence between diagonal Gaussians and the product space $(\mathbb{H}^2)^n$.

---

[1] diagonal or even spherical, for simplicity.

**Fisher distance, KL & Gaussian embeddings.** The above paragraph lets us relate the WORD2GAUSS algorithm (Vilnis & McCallum, 2015) to hyperbolic word embeddings. Although one could object that WORD2GAUSS is trained using a KL divergence, while hyperbolic embeddings relate to Gaussian distributions via the Fisher distance $d_F$, let us remind that KL and $d_F$ define the same local geometry. Indeed, the KL is known to be related to $d_F$, as its local approximation (Jeffreys, 1946). In short, if $P(\theta + d\theta)$ and $P(\theta)$ denote two closeby probability distributions for a small $d\theta$, then $\text{KL}(P(\theta + d\theta)||P(\theta)) = (1/2)\sum_{ij} g_{ij}d\theta^i d\theta^j + \mathcal{O}(\|d\theta\|^3)$, where $(g_{ij})_{ij}$ is the Fisher information metric, inducing $d_F$.

**Riemannian optimization.** A benefit of representing words in (products of) hyperbolic spaces, as opposed to (diagonal) Gaussian distributions, is that one can use recent Riemannian adaptive optimization tools such as RADAGRAD (Bécigneul & Ganea, 2018). Note that without this connection, it would be unclear how to define a variant of ADAGRAD (Duchi et al., 2011) intrinsic to the statistical manifold of Gaussians. Empirically, we indeed noticed better results using RADAGRAD, rather than simply Riemannian SGD (Bonnabel, 2013). Similarly, note that GLOVE trains with ADAGRAD.

## 6 ANALOGIES FOR HYPERBOLIC/GAUSSIAN EMBEDDINGS

The connection exposed in section 5 allows us to provide mathematically grounded *(i)* analogy computations for Gaussian embeddings using hyperbolic geometry, and *(ii)* hypernymy detection for hyperbolic embeddings using Gaussian distributions.

A common task used to evaluate word embeddings, called *analogy*, consists in finding which word $d$ is to the word $c$, what the word $b$ is to the word $a$. For instance, *queen* is to *woman* what *king* is to *man*. In the Euclidean embedding space, the solution to this problem is usually taken geometrically as $d = c + (b - a) = b + (c - a)$. Note that the same $d$ is also to $b$, what $c$ is to $a$.

How should one intrinsically define "analogy parallelograms" in a space of Gaussian distributions? Note that Vilnis & McCallum (2015) do not evaluate their Gaussian embeddings on the analogy task, and that it would be unclear how to do so. However, since we can go back and forth between (diagonal) Gaussians and (products of) hyperbolic spaces as explained in section 5, we can use the fact that parallelograms are naturally defined in the Poincaré ball, by the notion of gyro-translation (Ungar, 2012, section 4). In the Poincaré ball, the two solutions $d_1 = c + (b-a)$ and $d_2 = b + (c-a)$ are respectively generalized to

$$d_1 = c \oplus gyr[c, \ominus a](\ominus a \oplus b), \quad \text{and} \quad d_2 = b \oplus gyr[b, \ominus a](\ominus a \oplus c). \tag{6}$$

The formulas for these operations are described in closed-forms in appendix C, and are easy to implement. The fact that $d_1$ and $d_2$ differ is due to the curvature of the space. For evaluation, we chose a point $m_{d_1 d_2}^t := d_1 \oplus ((-d_1 \oplus d_2) \otimes t)$ located on the geodesic between $d_1$ and $d_2$ for some $t \in [0, 1]$; if $t = 1/2$, this is called the *gyro-midpoint* and then $m_{d_1 d_2}^{0.5} = m_{d_2 d_1}^{0.5}$, which is at equal hyperbolic distance from $d_1$ as from $d_2$. We explain in appendix A.2 how to select $t$, and that continuously deforming the Poincaré ball to the Euclidean space (by sending its radius to infinity) lets these analogy computations recover their Euclidean counterparts, which is a nice sanity check.

## 7 TOWARDS A PRINCIPLED SCORE FOR ENTAILMENT/HYPERNYMY

We now use the connection explained in section 5 to introduce a novel principled score that can be applied on top of our unsupervised learned Poincaré Glove embeddings to address the task of hypernymy detection, i.e. to predict relations of type `is-a(v,w)` such as `is-a(dog, animal)`. For this purpose, we first explain how learned hyperbolic word embeddings are mapped to Gaussian embeddings, and subsequently we define our hypernymy score.

**Invariance of distance-based embeddings to isometric transformations.** The method of Nickel & Kiela (2017) uses a heuristic entailment score in order to predict whether $u$ *is-a* $v$, defined for $u, v \in \mathbb{D}^n$ as is-a$(u, v) := -(1 + \alpha(\|v\|_2 - \|u\|_2))d(u, v)$, based on the intuition that the Euclidean norm should encode generality/specificity of a concept/word. However, such a choice *is not intrinsic* to the hyperbolic space when the training loss involves only the distance function. We say that

*training is intrinsic to* $\mathbb{D}^n$, *i.e.* invariant to applying any isometric transformation $\varphi : \mathbb{D}^n \to \mathbb{D}^n$ to all word embeddings (such as hyperbolic translation). But their "is-a" score is not intrinsic, i.e. depends on the parametrization. For this reason, we argue that an *isometry* has to be found and fixed before using the trained word embeddings in any non-intrinsic manner, e.g. to define hypernymy scores. To discover it, we leverage the connection between hyperbolic and Gaussian embeddings as follows.

**Mapping hyperbolic embeddings to Gaussian embeddings via an *isometry*.**     For a 1D Gaussian $\mathcal{N}(\mu, \sigma^2)$ representing a concept, generality should be naturally encoded in the magnitude of $\sigma$. As shown in section 5, the space of Gaussians endorsed with the Fisher distance is naturally mapped to the hyperbolic upper half-plane $\mathbb{H}^2$, where the variance $\sigma$ corresponds to the (positive) second coordinate in $\mathbb{H}^2 = \mathbb{R} \times \mathbb{R}_+^*$. Moreover, as shown in section 3, $\mathbb{H}^2$ can be isometrically mapped to $\mathbb{D}^2$, where the second coordinate $\sigma \in \mathbb{R}_+^*$ corresponds to the open vertical segment $\{0\} \times (-1, 1)$ in $\mathbb{D}^2$. However, in $\mathbb{D}^2$, any (hyperbolic) translation or any rotation w.r.t. the origin is an isometry[2]. Hence, in order to map a word $x \in \mathbb{D}^2$ to a Gaussian $\mathcal{N}(\mu, \sigma^2)$ via $\mathbb{H}^2$, we first have to find the *correct isometry*. This transformation should align $\{0\} \times (-1, 1)$ with whichever geodesic in $\mathbb{D}^2$ encodes generality. For simplicity, we assume it is composed of a centering and a rotation operations in $\mathbb{D}^2$. Thus, we start by identifying two sets $\mathcal{G}$ and $\mathcal{S}$ of potentially generic and specific words, respectively. For the re-centering, we then compute the means $g$ and $s$ of $\mathcal{G}$ and $\mathcal{S}$ respectively, and $m := (s + g)/2$, and Möbius translate all words by the global mean with the operation $w \mapsto \ominus m \oplus w$. For the rotation, we set $u := (\ominus m \oplus g)/\| \ominus m \oplus g\|_2$, and rotate all words so that $u$ is mapped to $(0, 1)$. Figure 2 and Algorithm 1 illustrate these steps.

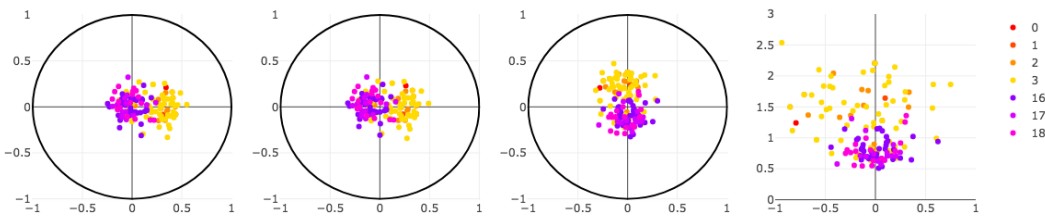

Figure 2: We show here one of the $\mathbb{D}^2$ spaces of 20D word embeddings embedded in $(\mathbb{D}^2)^{10}$ with our unsupervised hyperbolic GLOVE algorithm. This illustrates the three steps of applying the isometry. From left to right: the trained embeddings, raw; then after centering; then after rotation; finally after isometrically mapping them to $\mathbb{H}^2$ as explained in section 3. The isometry was obtained with the weakly-supervised method *WordNet 400 + 400*. Legend: WordNet levels (root is 0). Model: $h = (\cdot)^2$, full vocabulary of 190k words. More of these plots for other $\mathbb{D}^2$ spaces are shown in appendix A.3.

In order to identify the two sets $\mathcal{G}$ and $\mathcal{S}$, we propose the following two methods.

- **Unsupervised 5K+5K**: a fully unsupervised method. We first define a restricted vocabulary of the 50k most frequent words among the unrestricted one of 190k words, and rank them by frequency; we then define $\mathcal{G}$ as the 5k most frequent ones, and $\mathcal{S}$ as the 5k least frequent ones of the 50k vocabulary (to avoid extremely rare words which might have received less signal during training).

- **Weakly-supervised WN** $x+x$: a weakly-supervised method that uses words from the WordNet hierarchy. We define $\mathcal{G}$ as the top $x$ words from the top 4 levels of the WordNet hierarchy, and $\mathcal{S}$ as $x$ of the bottom words from the bottom 3 levels, randomly sampled in case of ties.

**Gaussian embeddings.**     Vilnis & McCallum (2015) propose using is-a$(P, Q) := -KL(P\|Q)$ for distributions $P, Q$, the argument being that a low $KL(P\|Q)$ indicates that we can encode $Q$ easily as $P$, implying that $Q$ entails $P$. However, we would like to mitigate this statement. Indeed, if $P = \mathcal{N}(\mu, \sigma)$ and $Q = \mathcal{N}(\mu, \sigma')$ are two 1D Gaussian distributions with same mean, then $KL(P\|Q) = z^2 - 1 - \log(z)$ where $z := \sigma/\sigma'$, which is not a monotonic function of $z$. *This breaks the idea that the magnitude of the variance should encode the generality/specificity of the concept.*

---

[2]See http://bulatov.org/math/1001 for intuitive animations describing hyperbolic isometries.

**Another entailment score for Gaussian embeddings.** What would constitute a good number for the variance's magnitude of a $n$-dimensional Gaussian distribution $\mathcal{N}(\mu, \Sigma)$? It is known that 95% of its mass is contained within a hyper-ellipsoid of volume $V_\Sigma = V_n \sqrt{\det(\Sigma)}$, where $V_n$ denotes the volume of a ball of radius 1 in $\mathbb{R}^n$. For simplicity, we propose dropping the dependence in $\mu$ and define a simple score is-a$(\Sigma, \Sigma') := \log(V'_\Sigma) - \log(V_\Sigma) = \sum_{i=1}^{n}(\log(\sigma'_i) - \log(\sigma_i))$. Note that using difference of logarithms has the benefit of removing the scaling constant $V_n$, and makes the entailment score invariant to a rescaling of the covariance matrices: is-a$(r\Sigma, r\Sigma') = $ is-a$(\Sigma, \Sigma'), \forall r > 0$.

To compute this is-a score between two hyperbolic word embeddings, we first map all word embeddings to Gaussians as explained above and, subsequently, apply the above proposed is-a score. Algorithm 1 illustrates these steps. Results are shown in section 9: Figure 4 and Tables 6, 7.

---

**Algorithm 1** is-a(v, w) hypernymy score using Poincaré embeddings

---

1: **procedure** IS-A_SCORE(v, w)
2:     **Input:** $v, w \in (\mathbb{D}^2)^p$ with $v = [v_1, ..., v_p], w = [w_1, ..., w_p], v_i, w_i \in \mathbb{D}^2$
3:     **Output:** is-a(v, w) lexical entailment score
4:     $\mathcal{G} \leftarrow$ *set of Poincaré embeddings of generic words*
5:     $\mathcal{S} \leftarrow$ *set of Poincaré embeddings of specific words*
6:     **for** i from 1 to $p$ **do**
        // Fixing the correct isometry.
7:         $g_i \leftarrow \text{mean}\{x_i | x \in \mathcal{G}\}$             // Euclidean mean of generic words
8:         $s_i \leftarrow \text{mean}\{y_i | y \in \mathcal{S}\}$             // Euclidean mean of specific words
9:         $m_i \leftarrow (g_i + s_i)/2$                 // Total mean
10:         $v'_i \leftarrow \ominus m_i \oplus v_i$            // Möbius translation by $m_i$
11:         $w'_i \leftarrow \ominus m_i \oplus w_i$          // Möbius translation by $m_i$
12:         $u_i \leftarrow (\ominus m_i \oplus g_i)/|| \ominus m_i \oplus g_i ||_2$   // Compute rotation vector
13:         $v''_i \leftarrow rotate(v'_i, u_i)$            // rotate $v'_i$
14:         $w''_i \leftarrow rotate(w'_i, u_i)$          // rotate $w'_i$
        // Convert from Poincaré disk coordinates to half-plane coordinates.
15:         $\tilde{v}_i \leftarrow \text{poincare2halfplane}(v''_i)$
16:         $\tilde{w}_i \leftarrow \text{poincare2halfplane}(w''_i)$
        // Convert half-plane coordinates to Gaussian parameters.
17:         $\mu_i^v \leftarrow \tilde{v}_{i1}/\sqrt{2}; \sigma_i^v \leftarrow \tilde{v}_{i2}$
18:         $\mu_i^w \leftarrow \tilde{w}_{i1}/\sqrt{2}; \sigma_i^w \leftarrow \tilde{w}_{i2}$
        **return** $\sum_{i=0}^{p}(\log(\sigma_i^v) - \log(\sigma_i^w))$

---

## 8   EMBEDDING SYMBOLIC DATA IN A CONTINUOUS SPACE WITH MATCHING HYPERBOLICITY

Why would we embed words in a hyperbolic space? Given some symbolic data, such as a vocabulary along with similarity measures between words − in our case, co-occurrence counts $X_{ij}$ − can we understand in a principled manner which geometry would represent it best? Choosing the right metric space to embed words can be understood as selecting the right inductive bias − an essential step.

**$\delta$-hyperbolicity.** A particular quantity of interest describing qualitative aspects of metric spaces is the $\delta$-*hyperbolicity* which we formally define in appendix B. This metric introduced by Gromov (1987) quantifies the tree-likeliness of a space. However, for various reasons discussed in appendix B, we used the *averaged* $\delta$-hyperbolicity, denoted $\delta_{avg}$, defined by Albert et al. (2014). Intuitively, a low $\delta_{avg}$ of a finite metric space characterizes that this space has an underlying hyperbolic geometry, *i.e.* an approximate tree-like structure, and that the hyperbolic space would be well suited to isometrically embed it. We also report the ratio $2 * \delta_{avg}/d_{avg}$ (invariant to metric scaling), where $d_{avg}$ is the average distance in the finite space, as suggested by Borassi et al. (2015), whose low value also characterizes the "hyperbolicness" of the space.

**Computing $\delta_{avg}$.** Since our methods are trained on a weighted graph of co-occurrences, it makes sense to look for the corresponding hyperbolicity $\delta_{avg}$ of this symbolic data. The lower this value,

the more hyperbolic is the underlying nature of the graph, thus indicating that the hyperbolic space should be preferred over the Euclidean space for embedding words. However, in order to do so, one needs to be provided with a distance $d(i, j)$ for each pair of words $(i, j)$, while our symbolic data is only made of similarity measures. Note that one cannot simply associate the value $-\log(P_{ij})$ to $d(i, j)$, as this quantity is not symmetric. Instead, inspired from Eq. (3), we associate to words $i, j$ the distance[3] $h(d(i, j)) := -\log(X_{ij}) + b_i + b_j \geq 0$ with the choice $b_i := \log(X_i)$, *i.e.*

$$d(i, j) := h^{-1}(\log((X_i X_j)/X_{ij})). \tag{7}$$

Table 1 shows values for different choices of $h$. The discrete metric spaces we obtained for our symbolic data of co-occurrences appear to have a very low hyperbolicity, *i.e.* to be very much "hyperbolic", which suggests to embed words in (products of) hyperbolic spaces. We report in section 9 empirical results for $h = (\cdot)^2$ and $h = \cosh^2$.

| $h(x)$ | $\log(x)$ | $x$ | $x^2$ | $\cosh(x)$ | $\cosh^2(x)$ | $\cosh^4(x)$ | $\cosh^5(x)$ | $\cosh^{10}(x)$ |
|---|---|---|---|---|---|---|---|---|
| $d_{avg}$ | 18950.4 | 18.9505 | 4.3465 | 3.68 | 2.3596 | 1.7918 | 1.6888 | 1.4947 |
| $\delta_{avg}$ | 8498.6 | 0.7685 | 0.088 | 0.0384 | 0.0167 | 0.0072 | 0.0056 | 0.0026 |
| $2\delta_{avg}/d_{avg}$ | 0.8969 | 0.0811 | 0.0405 | 0.0209 | 0.0142 | 0.0081 | 0.0066 | 0.0034 |

Table 1: average distances, $\delta$-hyperbolicities and ratios computed via sampling for the metrics induced by different $h$ functions, as defined in Eq. (7).

## 9 EXPERIMENTS: SIMILARITY, ANALOGY, ENTAILMENT

**Experimental setup.** We trained all models on a corpus provided by Levy & Goldberg (2014); Levy et al. (2015) used in other word embeddings related work. Corpus preprocessing is explained in the above references. The dataset has been obtained from an English Wikipedia dump and contains 1.4 billion tokens. Words appearing less than one hundred times in the corpus have been discarded, leaving $189,533$ unique tokens. The co-occurrence matrix contains approximately 700 millions non-zero entries, for a symmetric window size of 10. All models were trained for 50 epochs, and unless stated otherwise, on the full corpus of 189,533 word types. For certain experiments, we also trained the model on a restricted vocabulary of the $50,000$ most frequent words, which we specify by appending either "(190k)" or "(50k)" to the experiment's name in the table of results.

**Poincaré models, Euclidean baselines.** We report results for both 100D embeddings trained in a 100D Poincaré ball, and for 50x2D embeddings, which were trained in the Cartesian product of 50 2D Poincaré balls. Note that in the case of both models, one word will be represented by exactly 100 parameters. For the Poincaré models we employ both $h(x) = x^2$ and $h(x) = \cosh^2(x)$. All hyperbolic models were optimized with RADAGRAD (Bécigneul & Ganea, 2018) as explained in Sec. 5. On the tasks of similarity and analogy, we compare against a 100D Euclidean GloVe model which was trained using the hyperparameters suggested in the original GloVe paper (Pennington et al., 2014). The vanilla GloVe model was optimized using ADAGRAD (Duchi et al., 2011). For the Euclidean baseline as well as for models with $h(x) = x^2$ we used a learning rate of 0.05. For Poincaré models with $h(x) = \cosh^2(x)$ we used a learning rate of 0.01.

**The initialization trick.** We obtained improvement in the majority of the metrics when initializing our Poincaré model with pretrained parameters. These were obtained by first training the same model on the restricted (50k) vocabulary, and then using this model as an initialization for the full (190K) vocabulary. This will be referred to as the "initialization trick". For fairness, we also trained the vanilla (Euclidean) GloVe model in the same fashion.

**Similarity.** Word similarity is assessed using a number of well established benchmarks shown in Table 2. We summarize here our main results, but more extensive experiments (including in lower dimensions) are in Appendix A.1. We note that, with a single exception, our 100D and 50x2D models outperform the vanilla Glove baselines in all settings.

---

[3]One can replace $\log(x)$ with $\log(1 + x)$ to avoid computing the logarithm of zero.

Table 2: Word similarity results for 100-dimensional models. Highlighted: the **best** and the $2^{nd}$ best.

| Experiment name | RareWord | WordSim | SimLex | SimVerb | MC | RG |
|---|---|---|---|---|---|---|
| 100D Vanilla GloVe | 0.3798 | 0.5901 | 0.2963 | 0.1675 | 0.6524 | 0.6894 |
| 100D Vanilla GloVe w/ init trick | 0.3787 | 0.5668 | 0.2964 | 0.1639 | 0.6562 | 0.6757 |
| 100D Poincaré GloVe $h(x) = \cosh^2(x)$, w/ init trick | 0.4187 | 0.6209 | **0.3208** | **0.1915** | 0.7833 | 0.7578 |
| 50x2D Poincaré GloVe $h(x) = \cosh^2(x)$, w/ init trick | **0.4276** | **0.6234** | 0.3181 | 0.189 | **0.8046** | **0.7597** |
| 50x2D Poincaré GloVe $h(x) = x^2$, w/ init trick | 0.4104 | 0.5782 | 0.3022 | 0.1685 | 0.7655 | 0.728 |

Table 3: Nearest neighbors (in terms of Poincaré distance) for some words using our 100D hyperbolic embedding model.

| | |
|---|---|
| **sixties** | seventies, eighties, nineties, 60s, 70s, 1960s, 80s, 90s, 1980s, 1970s |
| **dance** | dancing, dances, music, singing, musical, performing, hip-hop, pop, folk, dancers |
| **daughter** | wife, married, mother, cousin, son, niece, granddaughter, husband, sister, eldest |
| **vapor** | vapour, refrigerant, liquid, condenses, supercooled, fluid, gaseous, gases, droplet |
| **ronaldo** | cristiano, ronaldinho, rivaldo, messi, zidane, romario, pele, zinedine, xavi, robinho |
| **mechanic** | electrician, fireman, machinist, welder, technician, builder, janitor, trainer, brakeman |
| **algebra** | algebras, homological, heyting, geometry, subalgebra, quaternion, calculus, mathematics, unital, algebraic |

**Analogy.** For word analogy, we evaluate on the Google benchmark (Mikolov et al., 2013a) and its two splits that contain semantic and syntactic analogy queries. We also use a benchmark by MSR that is also commonly employed in other word embedding works. For the Euclidean baselines we use 3COSADD (Levy et al., 2015). For our models, the solution $d$ to the problem "which $d$ is to $c$, what $b$ is to $a$" is selected as $m_{d_1 d_2}^t$, as described in section 6. In order to select the best $t$ without overfitting on the benchmark dataset, we used the same 2-fold cross-validation method used by (Levy et al., 2015, section 5.1) (see our Table 15) − which resulted in selecting $t = 0.3$. We report our main results in Table 4, and more extensive experiments in various settings (including in lower dimensions) in appendix A.2. We note that the vast majority of our models outperform the vanilla Glove baselines, with the 100D hyperbolic embeddings being the absolute best.

Table 4: Word analogy results for 100-dimensional models. Highlighted: the **best** and the $2^{nd}$ best.

| Experiment name | Method | SemGoogle | SynGoogle | Google | MSR |
|---|---|---|---|---|---|
| 100D Vanilla GloVe | 3COSADD | 0.6005 | 0.5869 | 0.5931 | 0.4868 |
| 100D Vanilla GloVe w/ init trick | 3COSADD | 0.6427 | 0.595 | 0.6167 | 0.4826 |
| 100D Poincaré GloVe $h(x) = \cosh^2(x)$, w/ init. trick | Cosine dist | **0.6641** | **0.6088** | **0.6339** | **0.4971** |
| 50x2D Poincaré GloVe $h(x) = x^2$, w/ init. trick | Poincaré dist | 0.6582 | 0.6066 | 0.6300 | 0.4672 |
| 50x2D Poincaré GloVe $h(x) = \cosh^2(x)$, w/ init. trick | Poincaré dist | 0.6048 | 0.6042 | 0.6045 | 0.4849 |

**Hypernymy evaluation.** For hypernymy evaluation we use the Hyperlex (Vulić et al., 2017) and WBLess (subset of BLess) (Baroni & Lenci, 2011) datasets. We classify all the methods in three categories depending on the supervision used for word embedding learning and for the hypernymy score, respectively. For Hyperlex we report results in Tab. 6 and use the baseline scores reported in (Nickel & Kiela, 2017; Vulić et al., 2017). For WBLess we report results in Tab. 7 and use the baseline scores reported in (Nguyen et al., 2017).

Table 5: Some words selected from the 100 nearest neighbors and ordered according to the hypernymy score function for a 50x2D hyperbolic embedding model using $h(x) = x^2$.

| | |
|---|---|
| **reptile** | amphibians, carnivore, crocodilian, fish-like, dinosaur, alligator, triceratops |
| **algebra** | mathematics, geometry, topology, relational, invertible, endomorphisms, quaternions |
| **music** | performance, composition, contemporary, rock, jazz, electroacoustic, trio |
| **feeling** | sense, perception, thoughts, impression, emotion, fear, shame, sorrow, joy |

**Hypernymy results discussion.** We first note that our fully unsupervised 50x2D, $h(x) = x^2$ model outperforms all its corresponding baselines setting a new state-of-the-art on unsupervised WBLESS accuracy and matching the previous state-of-the-art on unsupervised HyperLex Spearman correlation.

Second, once a small amount of weakly supervision is used for the hypernymy score, we obtain significant improvements as shown in the same tables and also in Fig. 4. We note that this weak supervision is only as a post-processing step (after word embeddings are trained) for identifying the best direction encoding the variance of the Gaussian distributions as described in Sec. 7. Moreover, it does not consist of hypernymy pairs, but only of 400 or 800 generic and specific sets of words from WordNet. Even so, our unsupervised learned embeddings are remarkably able to outperform all (except WN-Poincaré) supervised embedding learning baselines on HyperLex which have the great advantage of using the hypernymy pairs to train the word embeddings.

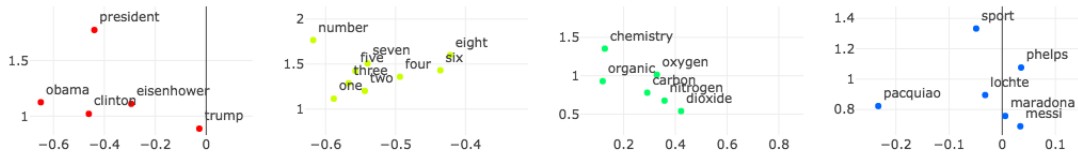

Figure 3: Different hierarchies captured by a 10x2D model with $h(x) = x^2$, in some selected 2D half-planes. The $y$ coordinate encodes the magnitude of the variance of the corresponding Gaussian embeddings, representing word generality/specificity. Thus, this type of Nx2D models offer an amount of interpretability.

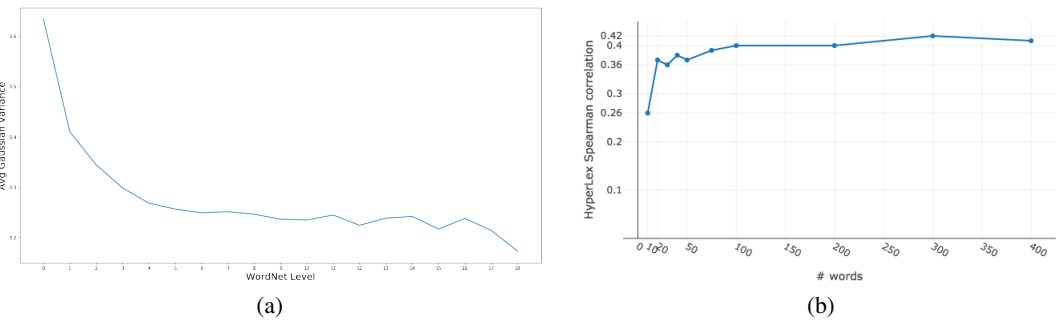

         (a)                                                 (b)

Figure 4: (4a): This plot describes how the Gaussian variances of our learned hyperbolic embeddings (trained unsupervised on co-occurrence statistics, isometry found with "Unsupervised 1k+1k") correlate with WordNet levels; (4b): This plot shows how the performance of our embeddings on hypernymy (HyperLex dataset) evolve when we increase the amount of supervision $x$ used to find the correct isometry in the model WN $x + x$. As can be seen, a very small amount of supervision (*e.g.* 20 words from WordNet) can significantly boost performance compared to fully unsupervised methods.

**Which model to choose?** While there is no single model that outperforms all the baselines on all presented tasks, one can remark that the model *50x2D, $h(x) = x^2$, with the initialization trick* obtains state-of-the-art results on hypernymy detection and is close to the best models for similarity and analogy (also Poincaré Glove models), but almost constantly outperforming the vanilla Glove baseline on these. This is the first model that can achieve competitive results on all these three tasks, additionally offering interpretability via the connection to Gaussian word embeddings.

Table 6: Hyperlex results in terms of Spearman correlation for different model types ordered according to their difficulty.

| MODEL TYPE | Method | $\rho$ |
|---|---|---|
| **Supervised** embedding learning & **Unsupervised** hypernymy score | OrderEmb | 0.191 |
| | PARAGRAM + CF | 0.320 |
| | WN-Basic | 0.240 |
| | WN-WuP | 0.214 |
| | WN-LCh | 0.214 |
| | WN-Eucl from (Nickel & Kiela, 2017) | 0.389 |
| | WN-Poincaré from (Nickel & Kiela, 2017) | **0.512** |
| **Unsupervised** embedding learning & **Weakly-supervised** hypernymy score | 50x2D Poincaré GloVe, $h(x) = \cosh^2(x)$, init trick (190k) | |
| | • WordNet 20+20 | 0.360 |
| | • WordNet 400+400 | 0.402 |
| | 50x2D Poincaré GloVe, $h(x) = x^2$, init trick (190k) | |
| | • WordNet 20+20 | 0.344 |
| | • WordNet 400+400 | **0.421** |
| **Unsupervised** embedding learning & **Unsupervised** hypernymy score | Word2Gauss-DistPos | 0.206 |
| | SGNS-Deps | 0.205 |
| | Frequency | 0.279 |
| | SLQS-Slim | 0.229 |
| | Vis-ID | 0.253 |
| | DIVE-W$\Delta$S (Chang et al., 2018) | 0.333 |
| | SBOW-PPMI-C$\Delta$S from (Chang et al., 2018) | **0.345** |
| | 50x2D Poincaré GloVe, $h(x) = \cosh^2(x)$, init trick (190k) 
 • Unsupervised 5k+5k | 0.284 |
| | 50x2D Poincaré GloVe, $h(x) = x^2$, init trick (190k) 
 • Unsupervised 5k+5k | **0.341** |

Table 7: WBLESS results in terms of accuracy for different model types ordered according to their difficulty.

| MODEL TYPE | Method | ACC. |
|---|---|---|
| **Supervised** embedding learning & **Unsupervised** hypernymy score | (Weeds et al., 2014) | 0.75 |
| | WN-Poincaré from (Nickel & Kiela, 2017) | 0.86 |
| | (Nguyen et al., 2017) | **0.87** |
| **Unsupervised** embedding learning & **Weakly-supervised** hypernymy score | 50x2D Poincaré GloVe, $h(x) = \cosh^2(x)$, init trick (190k) | |
| | • WordNet 20+20 | 0.728 |
| | • WordNet 400+400 | 0.749 |
| | 50x2D Poincaré GloVe, $h(x) = x^2$, init trick (190k) | |
| | • WordNet 20+20 | 0.781 |
| | • WordNet 400+400 | **0.790** |
| **Unsupervised** embedding learning & **Unsupervised** hypernymy score | SGNS from (Nguyen et al., 2017) | 0.48 |
| | (Weeds et al., 2014) | 0.58 |
| | 50x2D Poincaré GloVe, $h(x) = \cosh^2(x)$, init trick (190k) 
 • Unsupervised 5k+5k | 0.575 |
| | 50x2D Poincaré GloVe, $h(x) = x^2$, init trick (190k) 
 • Unsupervised 5k+5k | **0.652** |

## 10 CONCLUSION

We propose to adapt the GloVe algorithm to hyperbolic spaces and to leverage a connection between statistical manifolds of Gaussian distributions and hyperbolic geometry, in order to better interpret entailment relations between hyperbolic embeddings. We justify the choice of products of hyperbolic spaces via this connection to Gaussian distributions and via computations of the hyperbolicity of the symbolic data upon which GloVe is based. Empirically we present the first model that can simultaneously obtain state-of-the-art results or close on the three tasks of word similarity, analogy and hypernymy detection.

Our code is publicly available[4].

---

[4]https://github.com/alex-tifrea/poincare_glove

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

## A  MORE EXPERIMENTS

We show here extensive empirical results in various settings, including lower dimensions, different product structures, changing the vocabulary and using different $h$ functions.

**Experimental setup.**   In the experiment's name, we first indicate which dimension was used: "$nD$" denotes $\mathbb{D}^n$ while "$p \times kD$" denotes $(\mathbb{D}^k)^p$. "Vanilla" designates the baseline, *i.e.* the standard Euclidean GloVe from Eq. (1), while "Poincaré" designates our hyperbolic GloVe from Eq. (3). For Poincaré models, we then append to the experiment's name which $h$ function was applied to distances during training. Every model was trained for 50 epochs. Vanilla models were optimized with Adagrad (Duchi et al., 2011) while Poincaré models were optimized with RADAGRAD (Bécigneul & Ganea, 2018). For each experiment we tried using learning rates in $\{0.01, 0.05\}$, and found that the best were 0.01 for $h = \cosh^2$ and 0.05 for $h = (\cdot)^2$ and for Vanilla models − accordingly, we only report the best results. For similarity, we only considered the "target word vector" and always ignored the "context word vector". We also tried using the Euclidean/Möbius average[5] of these, but obtained (almost) consistently worse results for all experiments (including baselines) and do not report them.

### A.1  SIMILARITY

Reported scores are Spearman's correlations on the ranks for each benchmark dataset, as usual in the literature. We used (minus) the Poincaré distance as a similarity measure to rank neighbors.

Table 8: Unrestricted (190k) similarity results: models were trained and evaluated on the unrestricted (190k) vocabulary − "(init)" refers to the fact that the model was initialized with its counterpart (*i.e.* with same hyperparameters) on the restricted (50k) vocabulary, i.e. the *initialization trick*.

| Experiment's name | Rare Word | WordSim | SimLex | SimVerb | MC | RG |
|---|---|---|---|---|---|---|
| **100D Glove** | | | | | | |
| 100D Vanilla | 0.3798 | 0.5901 | 0.2963 | 0.1675 | 0.6524 | 0.6894 |
| 100D Vanilla (init) | 0.3787 | 0.5668 | 0.2964 | 0.1639 | 0.6562 | 0.6757 |
| 100D Poincaré, $\cosh^2$ | 0.3996 | **0.6486** | 0.3141 | 0.1777 | 0.7650 | 0.6834 |
| 100D Poincaré, $\cosh^2$ (init) | **0.4187** | 0.6209 | **0.3208** | **0.1915** | **0.7833** | **0.7578** |
| 100D Poincaré, $(\cdot)^2$ | 0.3762 | 0.4851 | 0.2732 | 0.1563 | 0.6540 | 0.6024 |
| 50x2D Poincaré, $\cosh^2$ | 0.4111 | 0.6367 | 0.3084 | 0.1811 | 0.7748 | 0.7250 |
| 50x2D Poincaré, $(\cdot)^2$ | 0.4106 | 0.5844 | 0.3007 | 0.1725 | 0.7586 | 0.7236 |
| | | | | | | |
| **48D Glove** | | | | | | |
| 48D Vanilla | 0.3497 | 0.5426 | 0.2525 | 0.1461 | 0.6213 | 0.6002 |
| 48D Poincaré, $\cosh^2$ | 0.3661 | **0.6040** | 0.2661 | **0.1539** | 0.7067 | 0.6466 |
| 48D Poincaré, $(\cdot)^2$ | 0.3574 | 0.4751 | 0.2418 | 0.1451 | 0.6780 | 0.6406 |
| 24x2D Poincaré, $\cosh^2$ | 0.3733 | 0.6020 | **0.2678** | 0.1513 | 0.7210 | 0.6595 |
| 24x2D Poincaré, $(\cdot)^2$ | **0.3825** | 0.5636 | 0.2655 | 0.1536 | **0.7857** | **0.6959** |
| | | | | | | |
| **20D Glove** | | | | | | |
| 20D Vanilla | 0.2930 | 0.4412 | 0.2153 | 0.1153 | 0.5120 | 0.5367 |
| 20D Poincaré, $\cosh^2$ | 0.3139 | 0.4672 | 0.2147 | 0.1226 | 0.5372 | 0.5720 |
| 20D Poincaré, $(\cdot)^2$ | 0.3250 | 0.4227 | 0.1994 | 0.1182 | 0.5970 | 0.6188 |
| 10x2D Poincaré, $\cosh^2$ | 0.3239 | **0.4818** | **0.2329** | **0.1281** | 0.6028 | 0.5986 |
| 10x2D Poincaré, $(\cdot)^2$ | **0.3380** | 0.4785 | 0.2314 | 0.1239 | **0.6242** | **0.6349** |
| | | | | | | |
| **4D Glove** | | | | | | |
| 4D Vanilla | 0.1445 | 0.1947 | 0.1356 | 0.0602 | 0.2701 | 0.3403 |
| 4D Poincaré, $\cosh^2$ | 0.1901 | 0.2424 | **0.1432** | 0.0581 | 0.2299 | 0.3065 |
| 4D Poincaré, $(\cdot)^2$ | **0.2031** | **0.2782** | 0.1395 | **0.0612** | **0.3173** | **0.3626** |

---

[5]A Möbius average is a gyro-midpoint, as explained in section 6.

Table 9: Restricted (50k) similarity results: models were trained and evaluated on the restricted (50k) vocabulary − except for the "Vanilla (190k)" baseline, which was trained on the unrestricted (190k) vocabulary and evaluated on the restricted vocabulary.

| Experiment's name | Rare Word | WordSim | SimLex | SimVerb | MC | RG |
|---|---|---|---|---|---|---|
| **100D Glove** | | | | | | |
| 100D Vanilla (190k) | 0.4443 | 0.5986 | 0.3071 | 0.1705 | 0.7245 | 0.7114 |
| 100D Vanilla | 0.4512 | 0.6091 | 0.2913 | 0.1742 | 0.6881 | 0.7148 |
| 100D Poincaré, $\cosh^2$ | 0.4606 | **0.6577** | **0.3156** | 0.1987 | 0.7916 | 0.7382 |
| 100D Poincaré, $(\cdot)^2$ | 0.4183 | 0.5241 | 0.2792 | 0.1671 | 0.6975 | 0.6753 |
| 50x2D Poincaré, $\cosh^2$ | **0.4661** | 0.6510 | 0.3152 | **0.2033** | **0.8098** | 0.7705 |
| 50x2D Poincaré, $(\cdot)^2$ | 0.4444 | 0.6009 | 0.3038 | 0.1858 | 0.7963 | **0.7862** |
| | | | | | | |
| **48D Glove** | | | | | | |
| 48D Vanilla | **0.4299** | **0.6171** | 0.2777 | 0.1641 | 0.7262 | 0.6739 |
| 48D Poincaré, $\cosh^2$ | 0.4191 | 0.6070 | 0.2682 | **0.1694** | 0.7566 | 0.6973 |
| 48D Poincaré, $(\cdot)^2$ | 0.3808 | 0.4940 | 0.2449 | 0.1607 | 0.7334 | 0.6982 |
| 24x2D Poincaré, $\cosh^2$ | 0.4235 | 0.6044 | **0.2790** | 0.1636 | 0.7834 | 0.7294 |
| 24x2D Poincaré, $(\cdot)^2$ | 0.4121 | 0.5759 | 0.2703 | 0.1601 | **0.7911** | **0.7302** |
| | | | | | | |
| **20D Glove** | | | | | | |
| 20D Vanilla | 0.3695 | **0.5198** | 0.2426 | 0.1271 | 0.6683 | 0.5960 |
| 20D Poincaré, $\cosh^2$ | 0.3683 | 0.4913 | 0.2255 | 0.1317 | 0.6627 | 0.6384 |
| 20D Poincaré, $(\cdot)^2$ | 0.3355 | 0.4125 | 0.2100 | 0.1240 | 0.6603 | **0.6556** |
| 10x2D Poincaré, $\cosh^2$ | 0.3749 | 0.4893 | 0.2321 | 0.1254 | **0.6775** | 0.6367 |
| 10x2D Poincaré, $(\cdot)^2$ | **0.3771** | 0.4748 | **0.2438** | **0.1396** | 0.6502 | 0.6461 |
| | | | | | | |
| **4D Glove** | | | | | | |
| 4D Vanilla | 0.1744 | 0.2113 | 0.1470 | 0.0582 | 0.3227 | 0.3973 |
| 4D Poincaré, $\cosh^2$ | 0.2183 | **0.2799** | **0.1530** | **0.0745** | **0.4104** | **0.4548** |
| 4D Poincaré, $(\cdot)^2$ | **0.2265** | 0.2357 | 0.1273 | 0.0605 | 0.2784 | 0.3495 |

**Remark.** Note that restricting the vocabulary incurs a loss of certain pairs of words from the benchmark similarity datasets, hence similarity results on the restricted (50k) vocabulary from Table 9 should be analyzed with caution, and in the light of Tables 10 and 11 (especially for Rare Word).

Table 10: Percentage of word pairs that are dropped when replacing the unrestricted vocabulary of 190k words with the restricted one of the 50k most frequent words.

| | Rare Word | WordSim | SimLex | SimVerb | MC | RG |
|---|---|---|---|---|---|---|
| % | 67.55 | 0.84 | 1.00 | 9.85 | 3.33 | 6.15 |

Table 11: Initial number of word pairs in each benchmark similarity dataset.

| | Rare Word | WordSim | SimLex | SimVerb | MC | RG |
|---|---|---|---|---|---|---|
| # | 2034 | 353 | 999 | 3500 | 30 | 65 |

## A.2 ANALOGY

**Details and notations.** In the column "method", "3.c.a" denotes using 3COSADD to solve analogies, which was used for all Euclidean baselines; for Poincaré models, as explained in section 9, the solution to the analogy problem is computed as $m_{d_1 d_2}^t$ with $t = 0.3$, and then the nearest neighbor in the vocabulary is selected either with the Poincaré distance on the corresponding space, which we denote as "$d$", or with cosine similarity on the full vector, which we denote as "cos". Finally, note that each cell contains two numbers, designated by $w$ and $w + \tilde{w}$ respectively: $w$ denotes ignoring the context vectors, while $w + \tilde{w}$ denotes considering as our embeddings the Euclidean/Möbius average between the target vector $w$ and the context vector $\tilde{w}$. In each dimension, we bold best results for $w$.

Table 12: Unrestricted (190k) analogy results: models were trained and evaluated on the unrestricted (190k) vocabulary − "(init)" refers to the fact that the model was initialized with its counterpart (*i.e.* with same hyperparameters) on the restricted (50k) vocabulary, i.e. the *initialization trick*.

| Experiment's name | Method | Semantic Google analogy accuracy using $w/w + \tilde{w}$ | Syntactic Google analogy accuracy using $w/w + \tilde{w}$ | Total Google analogy accuracy using $w/w + \tilde{w}$ | MSR analogy accuracy using $w/w + \tilde{w}$ |
|---|---|---|---|---|---|
| **100D Glove** | | | | | |
| 100D Vanilla | 3.c.a | 0.6005 / 0.6374 | 0.5869 / 0.5540 | 0.5931 / 0.5918 | 0.4868 / 0.4427 |
| 100D Vanilla (init) | 3.c.a | 0.6427 / 0.6878 | 0.5950 / 0.5672 | 0.6167 / 0.6219 | 0.4826 / 0.4509 |
| 100D Poincaré, $\cosh^2$ | $d$ | 0.4289 / 0.4444 | 0.5892 / 0.5484 | 0.5165 / 0.5012 | 0.4625 / 0.4186 |
| | cos | 0.4834 / 0.4908 | 0.5736 / 0.5514 | 0.5326 / 0.5239 | 0.4833 / 0.4395 |
| 100D Poincaré, $\cosh^2$ | $d$ | 0.6010 / 0.6308 | **0.6121** / 0.5659 | 0.6070 / 0.5954 | 0.4793 / 0.4375 |
| (init) | cos | **0.6641** / 0.6776 | 0.6088 / 0.5740 | **0.6339** / 0.6210 | **0.4971** / 0.4600 |
| 100D Poincaré, $(\cdot)^2$ | $d$ | 0.1013 / 0.5110 | 0.2388 / 0.4865 | 0.1764 / 0.4976 | 0.1461 / 0.3235 |
| | cos | 0.4329 / 0.7152 | 0.2507 / 0.4596 | 0.3334 / 0.5756 | 0.2042 / 0.3628 |
| 50x2D Poincaré, $\cosh^2$ | $d$ | 0.4511 / 0.4745 | 0.5766 / 0.5365 | 0.5196 / 0.5083 | 0.4763 / 0.4268 |
| | cos | 0.3274 / 0.3553 | 0.4326 / 0.3924 | 0.3849 / 0.3756 | 0.3329 / 0.2914 |
| 50x2D Poincaré, $(\cdot)^2$ | $d$ | 0.6426 / 0.6709 | 0.5940 / 0.5560 | 0.6160 / 0.6081 | 0.4576 / 0.4166 |
| | cos | 0.4754 / 0.5255 | 0.4544 / 0.4271 | 0.4639 / 0.4718 | 0.3425 / 0.2980 |
| | | | | | |
| **48D Glove** | | | | | |
| 48D Vanilla | 3.c.a | 0.3642 / 0.3650 | 0.451 / 0.4156 | 0.4115 / 0.3927 | 0.3467 / 0.3139 |
| 48D Poincaré, $\cosh^2$ | $d$ | 0.2368 / 0.2403 | 0.4693 / 0.4242 | 0.3638 / 0.3407 | 0.3755 / 0.3255 |
| | cos | 0.2479 / 0.2449 | 0.4704 / 0.4264 | 0.3694 / 0.3440 | **0.3919** / 0.3405 |
| 48D Poincaré, $(\cdot)^2$ | $d$ | 0.2108 / 0.4575 | 0.2752 / 0.4452 | 0.2460 / 0.4508 | 0.1842 / 0.2790 |
| | cos | 0.4513 / 0.5848 | 0.3137 / 0.4334 | 0.3762 / 0.5021 | 0.2386 / 0.3232 |
| 24x2D Poincaré, $\cosh^2$ | $d$ | 0.2338 / 0.2412 | 0.4509 / 0.4116 | 0.3524 / 0.3343 | 0.3426 / 0.3039 |
| | cos | 0.1294 / 0.1445 | 0.2240 / 0.1971 | 0.1811 / 0.1733 | 0.1619 / 0.1427 |
| 24x2D Poincaré, $(\cdot)^2$ | $d$ | **0.4663** / 0.4851 | **0.4834** / 0.4482 | **0.4756** / 0.4650 | 0.3456 / 0.3124 |
| | cos | 0.2479 / 0.2477 | 0.2626 / 0.2445 | 0.2559 / 0.2460 | 0.1670 / 0.1388 |
| | | | | | |
| **20D Glove** | | | | | |
| 20D Vanilla | 3.c.a | 0.1234 / 0.1202 | 0.2133 / 0.2004 | 0.1724 / 0.1640 | 0.1481 / 0.1281 |
| 20D Poincaré, $\cosh^2$ | $d$ | 0.1043 / 0.1020 | 0.2159 / 0.1946 | 0.1653 / 0.1526 | 0.1751 / 0.1527 |
| | cos | 0.1027 / 0.0993 | 0.2184 / 0.1955 | 0.1659 / 0.1519 | 0.1781 / 0.1505 |
| 20D Poincaré, $(\cdot)^2$ | $d$ | 0.1728 / 0.1840 | 0.2717 / 0.2646 | 0.2268 / 0.2280 | 0.1580 / 0.1451 |
| | cos | **0.2133** / 0.2018 | **0.2950** / 0.2762 | **0.2579** / 0.2424 | **0.1821** / 0.1611 |
| 10x2D Poincaré, $\cosh^2$ | $d$ | 0.1005 / 0.1015 | 0.2102 / 0.1915 | 0.1604 / 0.1506 | 0.1570 / 0.1365 |
| | cos | 0.0424 / 0.0392 | 0.0773 / 0.0686 | 0.0615 / 0.0553 | 0.0520 / 0.0446 |
| 10x2D Poincaré, $(\cdot)^2$ | $d$ | 0.1635 / 0.1618 | 0.2530 / 0.2263 | 0.2124 / 0.1970 | 0.1580 / 0.1446 |
| | cos | 0.0599 / 0.0548 | 0.0992 / 0.0861 | 0.0814 / 0.0719 | 0.0501 / 0.0408 |
| | | | | | |
| **4D Glove** | | | | | |
| 4D Vanilla | 3.c.a | 0.0036 / 0.0045 | 0.0012 / 0.0015 | 0.0023 / 0.0028 | 0.0011 / 0.0012 |
| 4D Poincaré, $\cosh^2$ | $d$ | 0.0089 / 0.0092 | 0.0043 / 0.0041 | 0.0064 / 0.0064 | 0.0046 / 0.0054 |
| | cos | 0.0036 / 0.0039 | 0.0020 / 0.0026 | 0.0027 / 0.0032 | 0.0015 / 0.0016 |
| 4D Poincaré, $(\cdot)^2$ | $d$ | **0.0135** / 0.0133 | **0.0058** / 0.0061 | **0.0093** / 0.0094 | **0.0051** / 0.0056 |
| | cos | 0.0045 / 0.0050 | 0.0024 / 0.0029 | 0.0034 / 0.0038 | 0.0015 / 0.0011 |

Table 13: Restricted (50k) analogy results: models were trained and evaluated on the restricted (50k) vocabulary − except for the "Vanilla (190k)" baseline, which was trained on the unrestricted (190k) vocabulary and evaluated on the restricted vocabulary.

| Experiment's name | Method | Semantic Google analogy accuracy using $w/w + \tilde{w}$ | Syntactic Google analogy accuracy using $w/w + \tilde{w}$ | Total Google analogy accuracy using $w/w + \tilde{w}$ | MSR analogy accuracy using $w/w + \tilde{w}$ |
|---|---|---|---|---|---|
| **100D Glove** | | | | | |
| 100D Vanilla (190k) | 3.c.a | 0.4789 / 0.4966 | 0.5684 / 0.5450 | 0.5278 / 0.5230 | 0.4382 / 0.3990 |
| 100D Vanilla | 3.c.a | 0.2848 / 0.3043 | 0.5003 / 0.5103 | 0.4025 / 0.4168 | 0.3545 / 0.3655 |
| 100D Poincaré, $\cosh^2$ | $d$ | 0.3684 / 0.3803 | **0.5820** / 0.5545 | 0.4851 / 0.4754 | 0.4394 / 0.3970 |
| | cos | 0.3982 / 0.4014 | 0.5786 / 0.5504 | 0.4968 / 0.4828 | **0.4494** / 0.4016 |
| 100D Poincaré, $(\cdot)^2$ | $d$ | 0.1265 / 0.4005 | 0.2209 / 0.4693 | 0.1781 / 0.4381 | 0.1384 / 0.3066 |
| | cos | 0.2634 / 0.5179 | 0.2521 / 0.4460 | 0.2572 / 0.4786 | 0.1933 / 0.3354 |
| 50x2D Poincaré, $\cosh^2$ | $d$ | 0.3956 / 0.4012 | 0.5799 / 0.5451 | 0.4963 / 0.4798 | 0.4482 / 0.3957 |
| | cos | 0.2809 / 0.2789 | 0.4146 / 0.4067 | 0.3539 / 0.3488 | 0.3464 / 0.2880 |
| 50x2D Poincaré, $(\cdot)^2$ | $d$ | **0.5204** / 0.5275 | 0.5819 / 0.5518 | **0.5540** / 0.5407 | 0.4404 / 0.3980 |
| | cos | 0.3873 / 0.4172 | 0.4517 / 0.4411 | 0.4225 / 0.4303 | 0.3335 / 0.2933 |
| | | | | | |
| **48D Glove** | | | | | |
| 48D Vanilla | 3.c.a | 0.3212 / 0.3299 | 0.4727 / 0.4303 | 0.4039 / 0.3847 | 0.3550 / 0.3156 |
| 48D Poincaré, $\cosh^2$ | $d$ | 0.2127 / 0.2163 | 0.4680 / 0.4239 | 0.3521 / 0.3297 | 0.3581 / 0.3078 |
| | cos | 0.2180 / 0.2220 | 0.4690 / 0.4228 | 0.3551 / 0.3317 | **0.3708** / 0.3134 |
| 48D Poincaré, $(\cdot)^2$ | $d$ | 0.2035 / 0.3676 | 0.2572 / 0.4129 | 0.2329 / 0.3923 | 0.1787 / 0.2652 |
| | cos | 0.3063 / 0.4212 | 0.3090 / 0.4174 | 0.3078 / 0.4192 | 0.2243 / 0.2951 |
| 24x2D Poincaré, $\cosh^2$ | $d$ | 0.2307 / 0.2308 | 0.4506 / 0.4090 | 0.3508 / 0.3281 | 0.3289 / 0.2979 |
| | cos | 0.1328 / 0.1334 | 0.2475 / 0.2153 | 0.1955 / 0.1781 | 0.1850 / 0.1544 |
| 24x2D Poincaré, $(\cdot)^2$ | $d$ | **0.3649** / 0.3788 | **0.4738** / 0.4343 | **0.4244** / 0.4091 | 0.3424 / 0.2985 |
| | cos | 0.2041 / 0.2080 | 0.2680 / 0.2469 | 0.2390 / 0.2293 | 0.1805 / 0.1611 |
| | | | | | |
| **20D Glove** | | | | | |
| 20D Vanilla | 3.c.a | 0.1223 / 0.1164 | 0.2472 / 0.2120 | 0.1905 / 0.1686 | 0.1550 / 0.1289 |
| 20D Poincaré, $\cosh^2$ | $d$ | 0.0925 / 0.0903 | 0.2292 / 0.1967 | 0.1672 / 0.1484 | 0.1601 / 0.1286 |
| | cos | 0.0917 / 0.0890 | 0.2355 / 0.1964 | 0.1702 / 0.1477 | 0.1629 / 0.1271 |
| 20D Poincaré, $(\cdot)^2$ | $d$ | 0.1583 / 0.1661 | 0.2619 / 0.2479 | 0.2149 / 0.2108 | 0.1624 / 0.1419 |
| | cos | **0.1757** / 0.1777 | **0.2970** / 0.2613 | **0.2408** / 0.2220 | **0.1804** / 0.1554 |
| 10x2D Poincaré, $\cosh^2$ | $d$ | 0.0962 / 0.0945 | 0.2177 / 0.1919 | 0.1626 / 0.1477 | 0.1546 / 0.1279 |
| | cos | 0.0403 / 0.0387 | 0.0850 / 0.0704 | 0.0647 / 0.0560 | 0.0483 / 0.0432 |
| 10x2D Poincaré, $(\cdot)^2$ | $d$ | 0.1440 / 0.1467 | 0.2533 / 0.2231 | 0.2037 / 0.1884 | 0.1580 / 0.1417 |
| | cos | 0.0614 / 0.0583 | 0.1014 / 0.0880 | 0.0832 / 0.0745 | 0.0473 / 0.0435 |
| | | | | | |
| **4D Glove** | | | | | |
| 4D Vanilla | 3.c.a | 0.0050 / 0.0053 | 0.0011 / 0.0012 | 0.0029 / 0.0031 | 0.0011 / 0.0011 |
| 4D Poincaré, $\cosh^2$ | $d$ | 0.0054 / 0.0056 | 0.0037 / 0.0037 | 0.0045 / 0.0046 | 0.0041 / 0.0044 |
| | cos | 0.0038 / 0.0036 | 0.0023 / 0.0025 | 0.0030 / 0.0030 | 0.0006 / 0.0006 |
| 4D Poincaré, $(\cdot)^2$ | $d$ | **0.0127** / 0.0127 | **0.0072** / 0.0077 | **0.0097** / 0.0100 | **0.0061** / 0.0057 |
| | cos | 0.0060 / 0.0061 | 0.0030 / 0.0037 | 0.0043 / 0.0048 | 0.0024 / 0.0025 |

**Remark.** Note that restricting the vocabulary to the most frequent 190k or 50k words will remove some of the test instances in the benchmark analogy datasets. These are described in Table 14.

Table 14: Number of test instances in the benchmark analogy datasets initially and after reductions to the vocabularies of the most frequent 190k and 50k words respectively.

|         | Semantic Google | Syntactic Google | Total Google | MSR  |
|---------|-----------------|------------------|--------------|------|
| initial | 8869            | 10675            | 19544        | 8000 |
| 190k    | 8649            | 10609            | 19258        | 7118 |
| 50k     | 6549            | 9765             | 16314        | 5778 |

Table 15: Result of the 2-fold cross-validation to determine which $t$ is best in $m^t_{d_1 d_2}$ (see section 6) to answer analogy queries. The (total) Google analogy dataset was randomly split in two partitions. For each partition, we selected the best $t$ across the 11 choices in $\{0, 0.1, 0.2, ..., 1\}$, and reported the test accuracy for this $t$. For both partitions, best results were obtained with $t = 0.3$.

|                                                    | Validation accuracy | Test accuracy |
|----------------------------------------------------|---------------------|---------------|
| Validation on partition 1
Test on partition 2   | 63.91               | 64.75         |
| Validation on partition 2
Test on partition 1   | 64.75               | 63.91         |

**About analogy computations.** Note that one can rewrite Eq. (6) with tools from differential geometry as

$$c \oplus gyr[c, \ominus a](\ominus a \oplus b) = \exp_c(P_{a \to c}(\log_a(b))), \tag{8}$$

where $P_{x \to y} = (\lambda_x / \lambda_y) gyr[y, \ominus x]$ denotes the *parallel transport* along the unique geodesic from $x$ to $y$. The $\exp$ and $\log$ maps of Riemannian geometry were related to the theory of gyrovector spaces (Ungar, 2008) by Ganea et al. (2018b), who also mention that when continuously deforming the hyperbolic space $\mathbb{D}^n$ into the Euclidean space $\mathbb{R}^n$, sending the curvature from $-1$ to $0$ (*i.e.* the radius of $\mathbb{D}^n$ from $1$ to $\infty$), the Möbius operations $\oplus^\kappa, \ominus^\kappa, \otimes^\kappa, gyr^\kappa$ recover their respective Euclidean counterparts $+, -, \cdot, Id$. Hence, the analogy solutions $d_1, d_2, m^t_{d_1 d_2}$ of Eq. (6) would then all recover $d = c + b - a$, which seems a nice sanity check.

## A.3   HYPERNYMY

We show here more plots illustrating the method (described in section 7) that we use to map points from a (product of) Poincaré disk(s) to a (diagonal) Gaussian. Colors indicate WordNet levels: low levels are closer to the root. Figures 5,6,7,8 show the three steps (centering, rotation, isometric mapping to half-plane) for 20D embeddings in $(\mathbb{D}^2)^{10}$, *i.e.* each of these steps in each of the 10 corresponding 2D spaces. In these figures, centering and rotation were determined with our proposed semi-supervised method, *i.e.* selecting 400+400 top and bottom words from the WordNet hierarchy. We show these plots for two models in $(\mathbb{D}^2)^{10}$: one trained with $h = (\cdot)^2$ and one with $h = \cosh^2$.

**Remark.** It is easily noticeable that words trained with $h = \cosh^2$ are embedded much closer to each other than those trained with $h = (\cdot)^2$. This is expected: $h$ is applied to the distance function, and according to Eq. (3), $d(w_i, \tilde{w}_j)$ should match $h^{-1}(b_i + \tilde{b}_j - \log(X_{ij}))$, which is smaller for $h = \cosh^2$ than for $h = (\cdot)^2$.

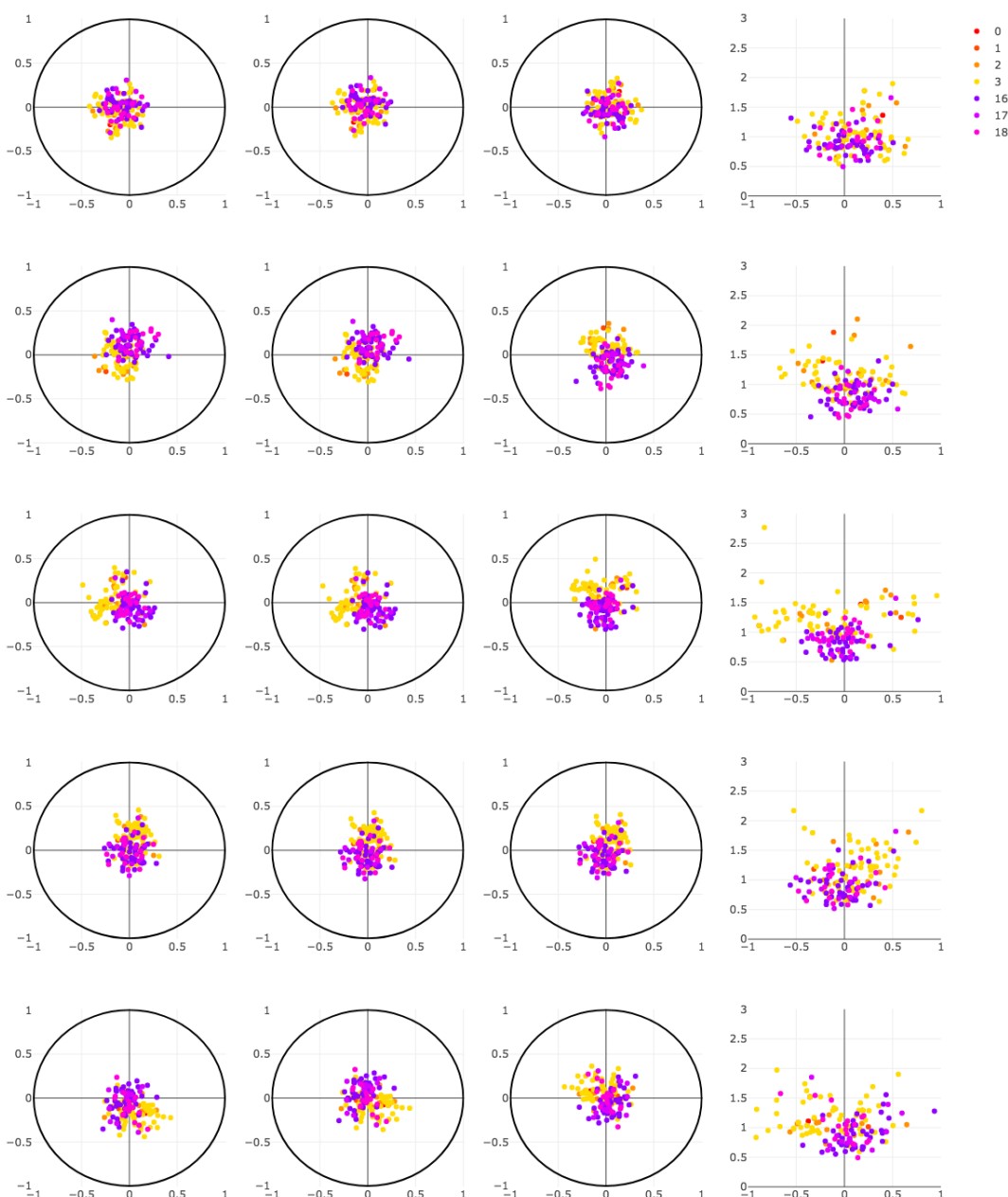

Figure 5: The first five 2D spaces of the model trained with $h = (\cdot)^2$.

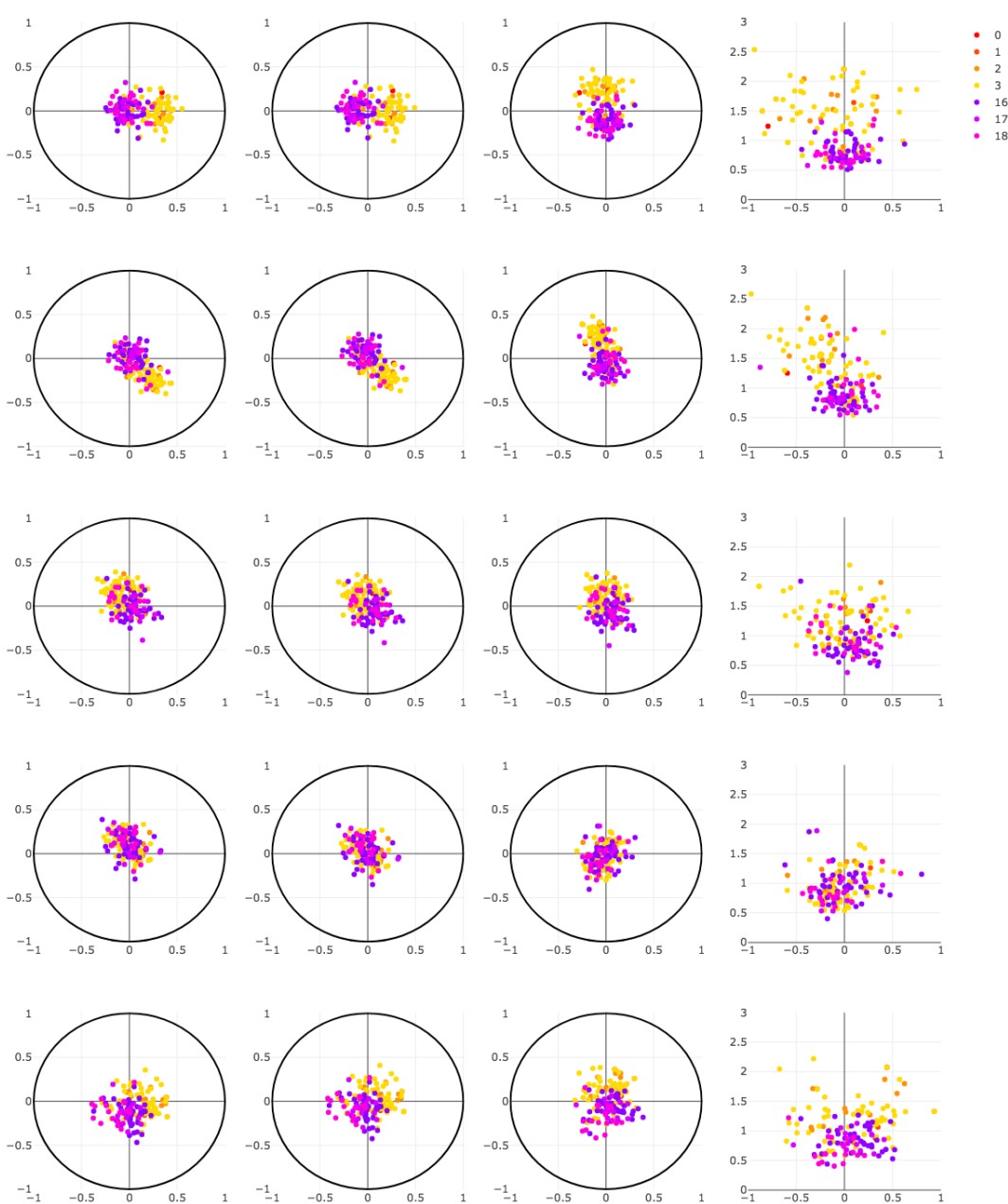

Figure 6: The last five 2D spaces of the model trained with $h = (\cdot)^2$.

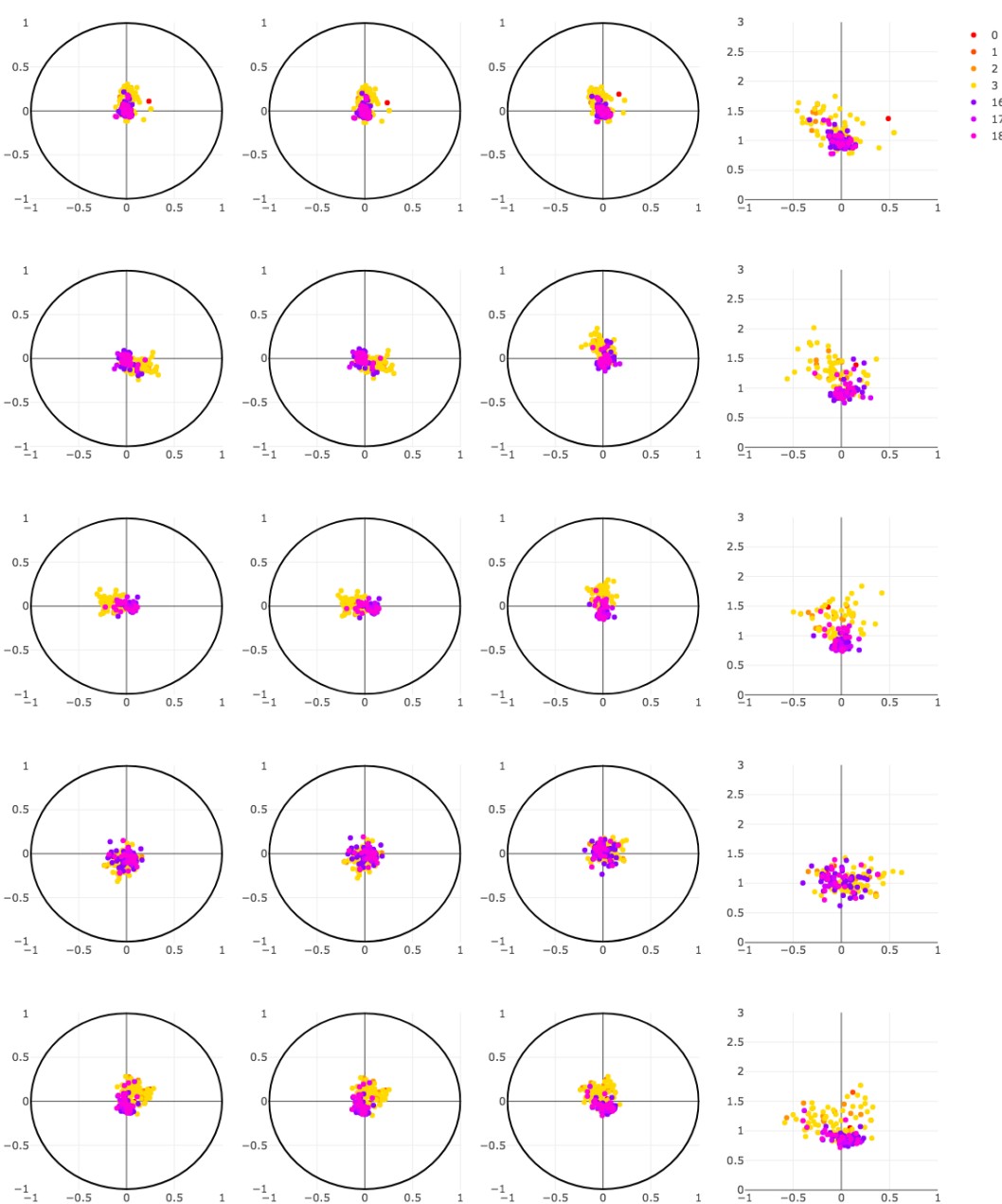

Figure 7: The first five 2D spaces of the model trained with $h = \cosh^2$.

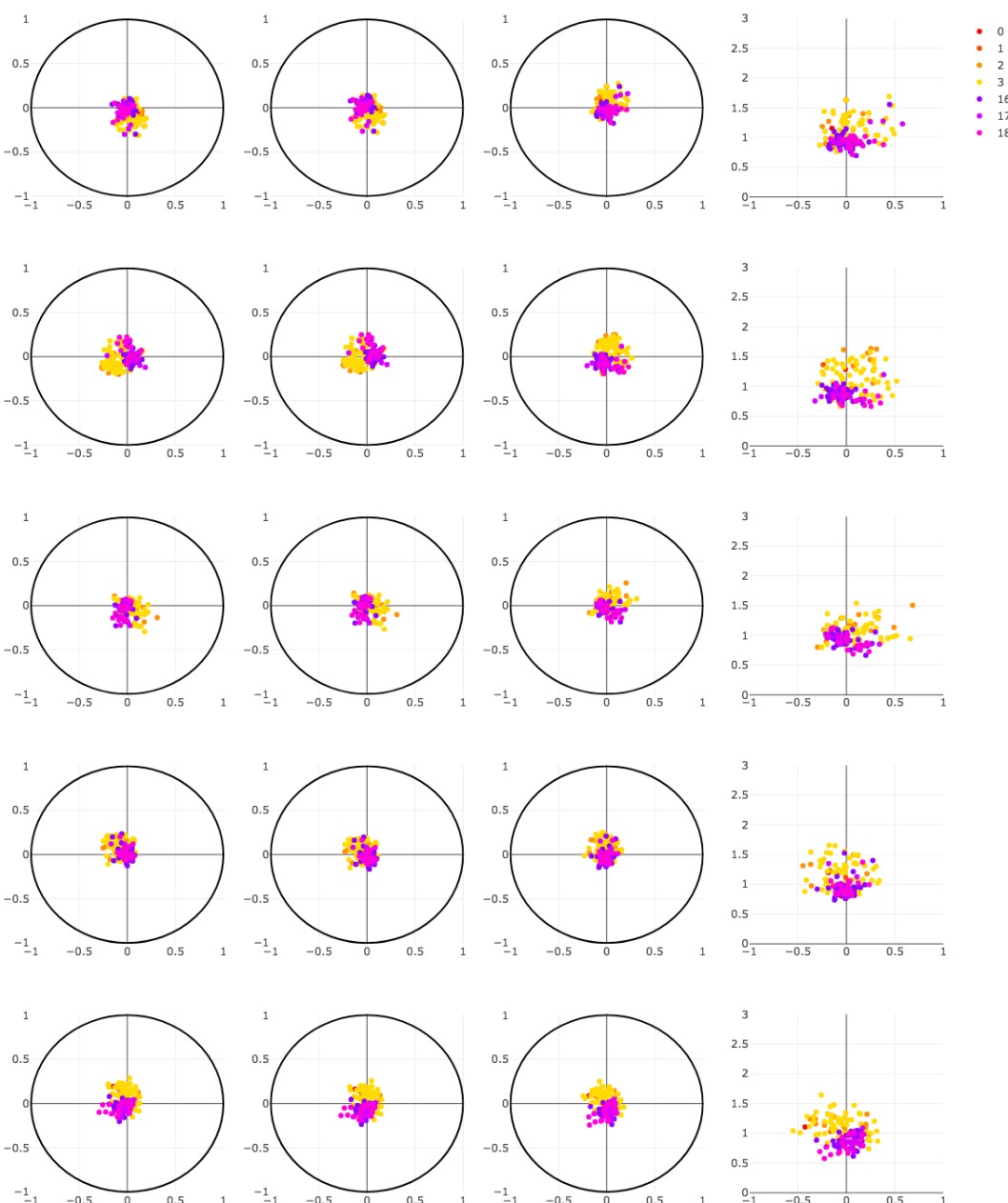

Figure 8: The last five 2D spaces of the model trained with $h = \cosh^2$.

# B  $\delta$-HYPERBOLICITY

Let us start by defining the $\delta$-*hyperbolicity*, introduced by Gromov (1987). The *hyperbolicity* $\delta(x, y, z, t)$ of a 4-tuple $(x, y, z, t)$ is defined as half the difference between the biggest two of the following sums: $d(x, y) + d(z, t)$, $d(x, z) + d(y, t)$, $d(x, t) + d(y, z)$. The $\delta$-hyperbolicity of a metric space is defined as the supremum of these numbers over all 4-tuples. Following Albert et al. (2014), we will denote this number by $\delta_{worst}$, and by $\delta_{avg}$ the average of these over all 4-tuples, when the space is a finite set. An equivalent and more intuitive definition holds for geodesic spaces, *i.e.* when we can define triangles: its $\delta$-hyperbolicity ($\delta_{worst}$) is the smallest $\delta > 0$ such that for any triangle $\Delta xyz$, there exists a point at distance at most $\delta$ from each side of the triangle. Chen et al. (2013) and Borassi et al. (2015) analyzed $\delta_{worst}$ and $\delta_{avg}$ for specific graphs, respectively. A low hyperbolicity of a graph indicates that it has an underlying hyperbolic geometry, *i.e.* that it is approximately tree-like, or at least that there exists a taxonomy of nodes. Conversely, a high hyperbolicity of a graph suggests that it possesses long cycles, or could not be embedded in a low dimensional hyperbolic space without distortion. For instance, the Euclidean space $\mathbb{R}^n$ is not $\delta$-hyperbolic for any $\delta > 0$, and is hence described as $\infty$-hyperbolic, while the Poincaré disk $\mathbb{D}^2$ is known to have a $\delta$-hyperbolicity of $\log(1 + \sqrt{2}) \simeq 0.88$. On the other-hand, a product $\mathbb{D}^2 \times \mathbb{D}^2$ is $\infty$-hyperbolic, because a $2D$ plane $\mathbb{R}^2$ could be isometrically embedded in it using for instance the first coordinates of each $\mathbb{D}^2$. However, if $\mathbb{D}^2$ would constitute a good choice to embed some given symbolic data, then most likely $\mathbb{D}^2 \times \mathbb{D}^2$ would as well. This stems from the fact that $\delta$-hyperbolicity ($\delta_{worst}$) is a worst case measure which does not reflect what one could call the "hyperbolic capacity" of the space. Furthermore, note that computing $\delta_{worst}$ requires $\mathcal{O}(n^4)$ for a graph of size $n$, while $\delta_{avg}$ can be approximated via sampling. Finally, $\delta_{avg}$ is robust to adding/removing a node from the graph, while $\delta_{worst}$ is not. For all these reasons, we choose $\delta_{avg}$ as a measure of hyperbolicity.

**More experiments.**  As explained in section 8, we computed hyperbolicities of the metric space induced by different $h$ functions, on the matrix of co-occurrence counts, as reported in Table 1. We also conducted similarity experiments, reported in Table 17. Apart from WordSim, results improved for higher powers of $\cosh$, corresponding to more hyperbolic spaces. However, also note that higher powers will tend to result in words embedded much closer to each other, *i.e.* with smaller distances, as explained in appendix A.3. In order to know whether this benefit comes from contracting distances or making the space more "hyperbolic", it would be interesting to learn (or cross-validate) the curvature $c$ of the Poincaré ball (or equivalently, its radius) jointly with the $h$ function. Finally, it order to explain why WordSim behaved differently compared to other benchmarks, we investigated different properties of these, as reported in Table 16. The geometry of the words appearing in WordSim do not seem to have a different hyperbolicity compared to other benchmarks; however, WordSim seems to contain much more frequent words. Since hyperbolicities are computed with the assumption that $b_i = \log(X_i)$ (see Eq. (7)), it would be interesting to explore whether learned biases indeed take these values. We left this as future work.

Table 16: Various properties of similarity benchmark datasets. The frequency index indicates the rank of a word in the vocabulary in terms of its frequency: a low index describes a frequent word. The median of indexes seems to best discriminate WordSim from SimLex and SimVerb.

| Property | WordSim | SimLex | SimVerb |
|---|---|---|---|
| # of test instances | 353 | 999 | 3,500 |
| # of different words | 419 | 1,027 | 822 |
| min. index (frequency) | 57 | 38 | 21 |
| max. index (frequency) | **58,286** | **128,143** | **180,417** |
| median of indexes | **2,723** | **4,463** | **9,338** |
| $\delta_{avg}, (\cdot)^2$ | 0.0738 | 0.0759 | 0.0799 |
| $\delta_{avg}, \cosh^2$ | 0.0154 | 0.0156 | 0.0164 |
| $2\delta_{avg}/d_{agv}, (\cdot)^2$ | 0.0381 | 0.0384 | 0.0399 |
| $2\delta_{avg}/d_{avg}, \cosh^2$ | 0.0136 | 0.0137 | 0.0143 |

Table 17: Similarity results on the unrestricted (190k) vocabulary for various $h$ functions. This table should be read together with Table 1.

| Experiment's name | Rare Word | WordSim | SimLex | SimVerb |
|---|---|---|---|---|
| 100D Vanilla | 0.3840 | 0.5849 | 0.3020 | 0.1674 |
| 100D Poincaré, $\cosh$ | 0.3353 | 0.5841 | 0.2607 | 0.1394 |
| 100D Poincaré, $\cosh^2$ | 0.3981 | **0.6509** | 0.3131 | 0.1757 |
| 100D Poincaré, $\cosh^3$ | 0.4170 | 0.6314 | 0.3155 | 0.1825 |
| 100D Poincaré, $\cosh^4$ | **0.4272** | 0.6294 | **0.3198** | **0.1845** |

## C  CLOSED-FORM FORMULAS OF MÖBIUS OPERATIONS

We show closed form expressions for the most common operations in the Poincaré ball, but we refer the reader to (Ungar, 2008; Ganea et al., 2018b) for more details.

**Möbius addition.**  The *Möbius addition* of $x$ and $y$ in $\mathbb{D}^n$ is defined as

$$x \oplus y := \frac{(1 + 2\langle x, y\rangle + \|y\|^2)x + (1 - \|x\|^2)y}{1 + 2\langle x, y\rangle + \|x\|^2\|y\|^2}. \tag{9}$$

We define $x \ominus y := x \oplus_c (-y)$.

**Möbius scalar multiplication.**  The *Möbius scalar multiplication* of $x \in \mathbb{D}^n \setminus \{\mathbf{0}\}$ by $r \in \mathbb{R}$ is defined as

$$r \otimes x := \tanh(r \tanh^{-1}(\|x\|))\frac{x}{\|x\|}, \tag{10}$$

and $r \otimes \mathbf{0} := \mathbf{0}$.

**Exponential and logarithmic maps.**  For any point $x \in \mathbb{D}^n$, the exponential map $\exp_x : T_x\mathbb{D}^n \to \mathbb{D}^n$ and the logarithmic map $\log_x : \mathbb{D}^n \to T_x\mathbb{D}^n$ are given for $v \neq \mathbf{0}$ and $y \neq x$ by:

$$\exp_x(v) = x \oplus \left(\tanh\left(\frac{\lambda_x\|v\|}{2}\right)\frac{v}{\|v\|}\right), \ \log_x(y) = \frac{2}{\lambda_x}\tanh^{-1}(\|-x \oplus y\|)\frac{-x \oplus y}{\|-x \oplus y\|}. \tag{11}$$

**Gyro operator and parallel transport.**  Parallel transport is given for $x, y \in \mathbb{D}^n, v \in T_x\mathbb{D}$ by the formula $P_{x \to y}(v) = \frac{\lambda_x}{\lambda_y} \cdot \text{gyr}[y, -x]v$. Gyr[6] is the gyroautomorphism on $\mathbb{D}^n$ with closed form expression shown in Eq. 1.27 of (Ungar, 2008):

$$\text{gyr}[u, v]w = \ominus(u \oplus v) \oplus \{u \oplus (v \oplus w)\} = w + 2\frac{Au + Bv}{D}. \tag{12}$$

where the quantities $A, B, D$ have closed-form expressions and are thus easy to implement:

$$A = -\langle u, w\rangle\|v\|^2 + \langle v, w\rangle + 2\langle u, v\rangle \cdot \langle v, w\rangle, \tag{13}$$

$$B = -\langle v, w\rangle\|u\|^2 - \langle u, w\rangle, \tag{14}$$

$$D = 1 + 2\langle u, v\rangle + \|u\|^2\|v\|^2. \tag{15}$$

---

[6]https://en.wikipedia.org/wiki/Gyrovector_space

