# OpenReview forum: "Poincare Glove: Hyperbolic Word Embeddings"
_ICLR.cc/2019/Conference_

### Official Review · AnonReviewer3 · 2018-10-26
**Quality in many respects.**

**Rating:** 7
**Confidence:** 3

**Review:**

The English, grammar and writing style is very good, as are the citations.
The technical quality appears to me to be very good (I am not an expert in Poincare spaces).
The authors demonstrate a good knowledge of the mathematical theory with the constructions made in Section 6.
The experimental write-up has been abbreviated.  The lexical entailment results Tables 6 and 7 are just sitting there without discussion, as far as I can see, as are the qualitative results Tables 4 and 5.   The entailment results are quite complex and really need supporting interpretation.  For instance, for Hyperlex, WN-Poincare is 0.512, above yours.
For your entailment score you say "For simplicity, we propose dropping the dependence in μ".  This needs more justification and discussion as it is counter-intuitive for those not expert in  Poincare spaces.
Section 6.2 presents the entailment score.  Note Nickel etal. give us a nice single formula.  You however, provide 4 paragraphs of construction from which an astute reader would then have to work on to extract your actual method.  I would prefer to see a summary algorithm given somewhere.  Perhaps you need another appendix.
RADAGRAD is discussed in Section 5, but I'd have preferred to see it discussed again in Section 8 and discussed to highlight what was indded done and the differences.  It certainly makes the paper non-reproducible.
A significant part of the theory in earlier sections is about the 50x2D method, but in experiments this doesn't seem to work as well.  Can you justify this some other how:  its much faster, its more interpretable?  Otherwise, I'm left thinking, why not delete this stuff?
The paper justifies its method with a substantial and winning comparison against vanilla GloVe.  That by itself is a substantial contribution.
But now, one is then hit with a raft of questions.  Embedding methods are popping up like daisies all over the fields of academia.  Indeed, word similarity and lexical entailment tasks themselves are proliferating too.  To me, its really unclear what one needs to achieve in the empirical section of a paper.  To make it worse, some folks use 500D, some 100D, some 50D, so results aren't always comparible.  Demonstrating one's work is state-of-the-art against all comers is a massive implementation effort.  I notice some papers now just compare against one other (e.g., Klami etal. ECML-PKDD, 2018).

My overall feeling is that this paper tries to compress too much into a small space (8 pages).
I think it really needs to be longer to present what is shown.   Moreover, I would want to see the inclusion of the work on 50x2D justified. So my criticisms are about the way the paper is written, not about the quality of the work.
Moroever, though, one needs to consider comparisons against models other than GloVe.

Addendum:  You know, what I really love about ICLR is the effort authors make to refresh their paper and respond to reviewers.  You guys did a great job.  Really impressed.  50x2D now clarified and some of the hasty/unexplained bits fixed.

---

> ### Author Response · Authors · 2018-11-13
> **Significantly improved how the paper is written**
>
> First, let us mention that we are happy to hear that our writing style was appreciated.
>
> We made significant improvements to the submission, which are listed in a general message in the thread. We tried to reply more specifically to your concerns below:
>
>
> Clarity of experimental results.
> You made a valid point saying that our experimental results needed better descriptions and interpretations. We updated this section accordingly. We hope that you will find the new version much clearer.
>
>
> Pseudo-code algorithm for computing entailment score.
> Thank you for this suggestion. We incorporated it, please see Algorithm 1 in section 7.
>
>
> WN-Poincaré is 0.512.
> This method is using word embeddings trained *supervised* using is-a relations, while ours is based on word embeddings trained *unsupervised* using raw text corpora. We only incorporated supervised baselines in the table to show that our unsupervised method manages to also outperform almost all supervised ones,  which is surprising. Thank you for pointing out that this was unclear. We hope that you will find our updated tables and experiment section clearer.
>
>
> 50x2D.
> Our new similarity and analogy tables have been updated by adding the "initialization trick" to the 50x2D model. As can be seen, the "init trick" significantly improves performance for this model on similarity and analogy. Moreover, this model achieves state-of-the-art results on unsupervised hypernymy detection (tables 6 and 7). So, the 50x2D model is competitive on all three tasks. Other reasons to preserve this model: (i) better interpretability, since once can visualize embeddings in each 2D space of the product; (ii) theoretically, 100D hyperbolic corresponds to 99D Gaussian with spherical variance (i.e. sigma^2 I), while 50x2D hyperbolic corresponds to a 50D Gaussian with a diagonal covariance, i.e. 50 variance parameters.
> These models allocate parameters in a different manner, and hence possess different strengths. This should be clearer in our revised version.
>
>
> Dropping dependence in mu.
> Heuristically, how general a concept is - when embedded as a Gaussian - should be encoded in the magnitude of its variance. Although the mean might also contain relevant information, discarding it makes the model simpler. Our empirical analysis shows that this model was sufficient to obtain state-of-the-art results among unsupervised methods on word-hypernymy. We leave further exploration of more complex models as future work.
>
>
> Radagrad.
> The Radagrad update is easy to implement, as described in Eq.(8) of [1]. We believe this should not compromise reproducibility. Moreover, upon acceptance, we would make our own implementation of Radagrad fully available, which should facilitate further research on this topic.
>
>
> Length of paper.
> We took into account your suggestion and used the full authorized length of 10 pages, to make the paper clearer. We hope that you will find this new write-up more comprehensible.
>
>
> [1] Riemannian adaptive optimization methods, Bécigneul & Ganea, arxiv.org/abs/1810.00760

---

### Official Review · AnonReviewer2 · 2018-10-30
**Adapting Glove word embedding to the Poincare half-plane: interesting but incremental**

**Rating:** 6
**Confidence:** 4

**Review:**

This paper adapts the Glove word embedding (Pennington et al 2014) to a hyperbolic space given by the Poincare half-plane model.  The embedding objective function is given by equation (3), where h=cosh^2 so that it corresponds to a hyperbolic geometry. The author(s) showed that their hyperbolic version of Glove is better than the original Glove. Besides that,  based on (Costa et al 2015), the author provided theoretical insights on the connection between hyperbolic embeddings with Gaussian word embeddings. Besides, the author(s) proposed a measure called "delta-hyperbolicity", that is based on (Gromov 1987) to study the model selection problem of using hyperbolic embeddings vs. traditional Euclidean embeddings.

Overall, I find the contributions are interesting but incremental. Therefore it may not be significant enough to be published in ICLR. Moreover, the experimental evaluation is insufficient to show the advantages of the proposed Poincare Glove model.

An interesting theoretical insight is that there exists an isometry between the Fisher-geodesic distance of diagonal Gaussians and a product of Poincare half-planes. This is interesting as it revealed a connection between hyperbolic embeddings with Gaussian embeddings, which is not widely known. However, this is not an original contribution. This connection is not related to the optimization of the proposed embedding, as Gaussian word embeddings are optimized based on KL divergence etc. that are easy to compute.

The computation of analogy based on isometric transformations is interesting but straightforward by applying translation operations in the Poincare ball. The novel contribution is minor and mainly on related empirical results.

The definition of the delta-hyperbolicity is missing. The explicit form of the definition should be clearly given in section 7. Again, this is not a novel contribution but an application of previous definitions (Gromov 1987).

In the word similarity and analogy experiments, the baseline is the vanilla Glove, this is not sufficient as it is widely known that hyperbolic embeddings can improve over Euclidean embeddings on certain datasets. The authors are therefore suggested to include another hyperbolic word embedding (e.g. Nickel and Kiela 2017) into the baselines and discuss the advantages and disadvantages of the proposed method.

There are no novel and well-abstracted theoretical results (theorems) given in the submission.

The length of the paper is longer than the recommended length (9 pages of main text).

---

> ### Author Response · Authors · 2018-11-13
> **Several contributions: one model strong in several tasks, a novel entailment score, state of the art unsupervised hypernymy results.**
>
> Thank you for your valuable comments. We understand that our initial presentation of experiments was suboptimal. We have updated this section. We are the first method to show competitive or state-of-the-art results simultaneously on the 3 tasks of word similarity, analogy and hypernymy detection (see also our reply for Reviewer1).
>
> After your comments, we also improved the presentation of our novel entailment score by updating section 7, in particular by introducing a pseudo-code description (see Algorithm 1).
>
>
> “Some presented mathematical notions are not novel”.
> Indeed, the definition of delta-hyperbolicity, the Fisher geometry of Gaussians being hyperbolic, and the definition of gyro-translation are not of our own. However, the combination of these notions into a new machine learning model for word embeddings and their usage for the construction of a completely new unsupervised hypernymy score allowed us to achieve high performance on different tasks with the same model, as well as state of the art on unsupervised word hypernymy detection (WBLESS results). We believe that achieving these results with the *same* model constitutes a valuable contribution.
>
>
> Analogy with gyro-translations.
> Indeed, the use of parallel/gyro-translations to solve the analogy task can be thought of as natural. However, as explained in section 6.1, because the space has non-zero curvature, there are two solutions to the analogy problem, which poses an unexpected difficulty. Our proposed solution described in sections 6, 9 and appendix A.2 is to select a point on the geodesic between these two solutions using a 2-fold cross-validation method. Lastly, let us note that the use of Euclidean translation is prohibited, as these operations belong to the ambient space, and their use would violate the hyperbolic structure.
>
>
> Delta-hyperbolicity.
> We gave the definition in the appendix. We chose not to include the definition in the main text, because we thought it would not improve the comprehension of the reader. Indeed, the intuition behind Gromov’s definition of the delta-hyperbolicity of a quadruple is relatively difficult to grasp. We would also like to draw your attention on the fact that we expanded this appendix with further experimental results, to better understand how hyperbolicity affects similarity results.
>
>
> Related work.
> Nickel & Kiela’s Poincaré Embeddings [1] is using word embeddings trained *supervised* using is-a relations, while ours is based on word embeddings trained *unsupervised* using raw text corpora. [1] evaluates on graph-reconstruction and link-prediction and hence only targets Word-hypernymy, and is not trained to perform well on Word-analogy or Word-similarity, which are tasks traditionally used to evaluate word embedding methods trained on raw text corpora.
>
>
> Optimization of Gaussian embeddings.
> As explained at the end of section 5, this connection allows us to use Riemannian Adagrad, which performs adaptivity across Poincare balls in the cartesian product. This optimization method is intrinsic to the statistical manifold of Gaussian distributions w.r.t. their Fisher geometry, and is hence both practically powerful and mathematically principled.
>
>
> Length of the paper.
> As suggested by Reviewer 3, in our updated version, we decided to use the full authorized length of 10 pages. The main reason for this is that we think our initial submission lacked clarity in certain places, especially in the way we presented our experimental results.  We also wanted to incorporate the modifications suggested by all reviewers. We hope that you will find this new form more convincing and a better fit to the conference.
>
>
> [1] Poincaré embeddings for learning hierarchical representations, Nickel & Kiela, NIPS 2017

---

### Official Review · AnonReviewer1 · 2018-11-06
**Poincare Glove: Hyperbolic Word Embeddings**

**Rating:** 6
**Confidence:** 4

**Review:**

Summary:
Words have implicit hierarchy among themselves in a text. Hyperbolic geometry due to the negative curvature and the delta-hyperbolicity is more suitable for representing hierarchical data in the continuous space. As a result it is natural to learn word representations/embeddings in the hyperbolic space. This paper proposes a promising approach that extends the approach presented in [1] to implement a GLOVE based hyperbolic word embedding model. The embeddings are optimized by using the Riemannian Optimization methods presented in [2]. Authors provide results on word-similarity and word-analogy tasks.


Questions:
What are the reasons for choosing a Poincare Ball model of the hyperbolic space instead of hyperboloid or other models of the hyperbolic space?
Can you expand on the role of gyr[.,.] in Equations 6 and 7.
Besides the tasks that are presented in this paper including word-analogy and the word-similarity tasks. Have you considered using the embeddings learned in hyperbolic space in a down-stream task such as NLI?

Pros:
The paper is very well-written, the motivation and the goals are quite clear.
The relationship between the Gaussian embeddings and the product spaces is interesting and neat. The paper is theoretically strong and consistent.
The idea of learning word-embeddings in hyperbolic space with the proposed approach is novel and relevant.

Cons:

The weakest point of this paper is the experiments. Unfortunately the results reported are underwhelming on WBLESS and the Hyperlex tasks compared to other published results. The paper presents convincing results on Word-analogy and Word-similarity tasks. However they do not compare against the published results on those tasks.

[1] Ganea, O. E., Bécigneul, G., & Hofmann, T. (2018). Hyperbolic Neural Networks. arXiv preprint arXiv:1805.09112.
[2] Bécigneul, Gary, and Octavian-Eugen Ganea. "Riemannian Adaptive Optimization Methods." arXiv preprint arXiv:1810.00760 (2018).

---

> ### Author Response · Authors · 2018-11-13
> **Our unsupervised model is the first to competitively tackle all 3 tasks of word hypernymy (SOTA on unsupervised WBLESS), similarity and analogy**
>
> Thank you for your positive feedback.
>
> Experiments section:
>
> We rephrased the Experiments section 9 to better describe our empirical results (see below).
>
>
> Hypernymy experiments:
>
> Our fully *unsupervised* method (unsupervised trained embeddings + unsupervised hypernymy score) obtains state-of-the-art (SOTA) on WBLESS and matches previous SOTA on Hyperlex tasks - see Tables 6 and 7.
>
> We also propose to use WordNet to progressively incorporate weak supervision into the hypernymy scoring function, but not into the word embedding training phase. This likely results in lower scores compared to methods that use hypernymy supervision for training embeddings. However, our models of type “unsupervised trained embeddings + weakly-supervised hypernymy score” outperform the vast majority of methods that use supervision at training time, which is very encouraging. And our only “weak supervision”  comes from 400+400=800 *word levels* of the WordNet hierarchy, without using any hypernymy relations per se.
>
> The separation between these 3 types of hypernymy detection methods was not clear in the original version of our paper, but should be in our updated version - please see Tables 6 and 7.
>
>
> Results on similarity and analogy:
>
> We did not compare against published results because state-of-the-art is currently held by GloVe trained on Wikipedia 2014 + Gigaword 5. We trained only on Wikipedia 2014, because we did not have access to Gigaword 5 due to its prohibitive cost. The size of the dataset makes a significant difference for GloVe, since this algorithm gathers co-occurrences, which are relatively noisy statistics. In future work, we might acquire this dataset and re-run experiments. For now, we believe that our baseline is fair, since both the Euclidean and hyperbolic methods are trained on the same dataset. Moreover, upon acceptance, we would make our code fully available, including evaluation scripts, which should facilitate further research on this topic.
>
>
> Questions:
>
> Poincaré ball: we chose this model because it was used by [1] and [2], but in future work it would be interesting to investigate whether other models would lead to better optimization. In particular, we plan to investigate using the Lorentz and half-plane models.
>
> The gyr operator: it is the rotational component of the parallel transport along geodesics, inherited from the curvature of the space. It casts the holonomy of the manifold into an linear map. It captures the default of commutativity of Mobius translations:  a \oplus b = gyr[a,b](b\oplus a), for all a,b in D^n. Although it is defined in the ball, it can be naturally extended to the ambient Euclidean space, which yields an isometry [3, remark 1.2 and Eq.(1.32)]. We provide pointers to the interested reader in the appendix.
>
> Downstream task: this is a very nice suggestion. We leave it as future work.
>
>
> [1] Poincaré embeddings for learning hierarchical representations, Nickel & Kiela, NIPS 2017
> [2] Hyperbolic neural networks, Ganea et al., NIPS 2018
> [3] A gyrovector space approach to hyperbolic geometry, Ungar A.

---

### Author Response · Authors · 2018-11-13
**Our experimental results and improvement of presentation/discussion**

First, we would like to warmly thank all three reviewers for their valuable comments, and for the time and effort they invested in understanding our work.

We took into consideration all your comments, and updated our submission accordingly and significantly (especially sections 7 and 9) to better emphasize your comments, our contributions and our empirical results.

In terms of experiments, our method achieves state-of-the-art (SOTA) on hypernymy detection on the WBLESS dataset for the class of fully end-to-end unsupervised methods, and matches SOTA on Hyperlex, at the same time simultaneously outperforming vanilla Glove on word similarity and analogy. If the reader is interested into on single model “good for all”, we analyze at the end of Section 9 the model “50x2D, with h(x)=x^2 and initialization trick”, which is competitive on all 3 tasks.

General paper modifications:

Main text:
-Rewriting of the entire section 7 to better explain the computation of the word entailment score.
-Pseudo-code algorithm to compute the entailment score (Algorithm 1).
-We rewrote the experiments section 9.
-Updated hypernymy tables (6,7) for better classification of SOTA baseline methods in various settings and better emphasis of our results as the unsupervised hypernymy SOTA.
-Updated similarity and analogy tables (2,4).
-Explanations and thorough discussions of results for the three tasks.
-New plots of Hyperlex performance w.r.t. the amount of WordNet supervision we incorporate for evaluation (figure 4).

Appendix:
-Four tables of extensive similarity and analogy results (tables 8,9,12,13).
-Plots of 20x2D embeddings (figures 5,6,7,8).
-Explanation of the midpoint selection procedure for solving analogies (table 15).
-Section on delta-hyperbolicity expanded with new similarity results (table 17).

More detailed responses are provided below each review.

---

### Meta-Review · Area_Chair1 · 2018-12-17
**Interesting submission**

**Confidence:** 5
**Recommendation:** Accept (Poster)

**Metareview:**

Word vectors are well studied but this paper adds yet another interesting dimension to the field.